# Mitochondrial flashes regulate ATP homeostasis in the heart

Xianhua Wang[1][*][†], Xing Zhang[2][†], Di Wu[1][†], Zhanglong Huang[1], Tingting Hou[1], Chongshu Jian[1], Peng Yu[1], Fujian Lu[1], Rufeng Zhang[1], Tao Sun[1], Jinghang Li[1], Wenfeng Qi[1], Yanru Wang[1], Feng Gao[2][*], Heping Cheng[1]

[1]State Key Laboratory of Membrane Biology, Beijing Key Laboratory of Cardiometabolic Molecular Medicine, Institute of Molecular Medicine, Peking-Tsinghua Center for Life Sciences, Peking University, Beijing, China; [2]Department of Aerospace Medicine, The Fourth Military Medical University, Xi'an, China

**Abstract** The maintenance of a constant ATP level ('set-point') is a vital homeostatic function shared by eukaryotic cells. In particular, mammalian myocardium exquisitely safeguards its ATP set-point despite 10-fold fluctuations in cardiac workload. However, the exact mechanisms underlying this regulation of ATP homeostasis remain elusive. Here we show mitochondrial flashes (mitoflashes), recently discovered dynamic activity of mitochondria, play an essential role for the auto-regulation of ATP set-point in the heart. Specifically, mitoflashes negatively regulate ATP production in isolated respiring mitochondria and, their activity waxes and wanes to counteract the ATP supply-demand imbalance caused by superfluous substrate and altered workload in cardiomyocytes. Moreover, manipulating mitoflash activity is sufficient to inversely shift the otherwise stable ATP set-point. Mechanistically, the Bcl-xL-regulated proton leakage through $F_1F_o$-ATP synthase appears to mediate the coupling between mitoflash production and ATP set-point regulation. These findings indicate mitoflashes appear to constitute a digital auto-regulator for ATP homeostasis in the heart.

**\*For correspondence:** xianhua@pku.edu.cn (XW); fgao@fmmu.edu.cn (FG)

[†]These authors contributed equally to this work

**Competing interests:** The authors declare that no competing interests exist.

## Introduction

ATP is the most important energy currency and mitochondria constitute the largest cellular power-house, supplying ATP at a rate of ~6 kg/day in the human heart (*Neubauer, 2007*). More than an energy currency, ATP also plays diverse roles in physiological processes including signal transduction, ion channel regulation, cytoskeleton remodelling, and gene transcription (*Rolfe and Brown, 1997*). The regulation of intracellular ATP concentration is therefore a vital homeostatic function shared by all tissues and eukaryotic cells. An exemplary case is found in mammalian myocardium - the rate of energy consumption and supply (i.e., ATP flux) augments by 5–10 folds in situations of fight-or-flight or exercise, but the ATP concentration (i.e., ATP set-point) remains remarkably constant (*Balaban et al., 1986*; *Neely et al., 1973*; *Matthews et al., 1981*; *Allue et al., 1996*). The mechanism underlying such exquisite ATP homeostatic regulation has been extensively pursued over the last several decades. ADP, $P_i$ and $Ca^{2+}$ have been suggested to be regulators of ATP homeostasis, but challenges exist because of the relative constancy of ADP and $P_i$ (*Katz et al., 1989*; *Yaniv et al., 2010*; *Balaban, 2012*) and the extremely low activity of the mitochondrial $Ca^{2+}$ uniporter (MCU) in cardiac myocytes (*Fieni et al., 2012*; *Williams et al., 2013*). Moreover, recent findings in MCU knockout mice suggest that MCU is dispensable for energy metabolism except under extreme physiological stress (*Pan et al., 2013*). Myocardial overexpression of the dominant negative MCU increases, rather than decreases, mitochondrial oxygen consumption due in part to secondary

**eLife digest** A small molecule called ATP is often referred to as the primary "energy currency" of living cells. It is required to power tasks as diverse as the general housekeeping processes that keep all cells alive to the programmed cell death response that dismantles any cells that are no longer needed. It is also crucial that cells maintain a constant level of ATP at all times, even when the supply of and demand for ATP fluctuate. This control is particularly important in the mammalian heart where the rates of ATP production and consumption change ten-fold during intense exercise. Despite intensive research over the past decades, it was still not known how cells keep ATP levels constant.

In many cell types, including heart muscle cells, ATP is mainly produced inside compartments called mitochondria. Each heart muscle cell contains between 5,000 and 8,000 mitochondria. Recent experiments have shown that ATP production in mitochondria is interrupted by ten-second bursts called "mitochondrial flashes" (or mitoflashes for short), during which the mitochondria release chemicals called reactive oxygen species. The mitoflashes are tightly linked with energy usage, and Wang, Zhang, Wu et al. have now explored if and how mitoflashes regulate ATP levels in the heart.

Experiments on isolated mitochondria from mouse heart muscle cells showed that mitoflashes inhibit the production of ATP. When the intact heart muscle cells were given excess of the building blocks needed to produce ATP – mitoflashes occurred more often. Conversely, when the cells were forced to contract more quickly, which increased demand for ATP, the mitoflashes occurred less often. Importantly, the level of ATP inside the cells actually remained constant in the experiments. Furthermore, inhibiting mitoflashes with antioxidants increased the ATP concentration in heart muscle cells. Lastly, Wang et al. demonstrated that mitoflashes could be triggered under certain conditions.

Overall, these experiments uncovered a way in which highly active cells can maintain a constant level of ATP. Future studies are needed to understand exactly how mitoflashes are initiated and how they in turn inhibit ATP production. A better understanding of these processes might uncover molecules that could be targeted by drugs to the control of the rate of ATP production to treat heart failure.

effect on cytosol $Ca^{2+}$ homeostasis (*Rasmussen et al., 2015*). To date, deciphering the exact mechanism underlying ATP set-point regulation in the heart remains an unmet challenge.

Mitochondrial flash (mitoflash) represents a recently discovered dynamic activity of the mitochondria (*Wang et al., 2008*) and has been ubiquitously detected in all eukaryotic species examined, from *C. elegans* to zebrafish and to rodents and humans (*Wang et al., 2008*; *Hou et al., 2014*; *Wang et al., 2016a*; *Shen et al., 2014*; *Zhang et al., 2015*). Individual mitoflash consists of multiple signal components including bursting superoxide production, transient matrix alkalization, oxidative redox shift, transient depletion of the electron donors NADH and $FADH_2$, and mitochondrial membrane potential depolarization (*Wang et al., 2008*, *2016b*). The generation of mitoflashes in intact cells requires the integrity of the electron transfer chain (ETC) (*Wang et al., 2008*); and mitoflash frequency is highly regulated over a wide dynamic range by factors including metabolic state, thus the mitoflash activity is considered a biomarker for mitochondrial energy metabolism under certain conditions (*Wei et al., 2011*; *Pouvreau, 2010*; *Fang et al., 2011*; *Gong et al., 2015*). Most recently, we have shown that protons produced by photolysis or electroneutral proton ionophores act as a powerful mitoflash trigger (*Wang et al., 2016b*). This finding is instructive because, in the Mitchell chemiosmotic theory of ATP synthesis (*Nicholls and Ferguson, 2002*; *Mitchell, 1961*), proton gradients, vectorial proton movement, and proton motive force ($\Delta\mu_H$) across the inner mitochondrial membrane are quintessential for energy metabolism. Thus, the mitoflash biogenesis may be mechanistically and functionally intertwined with energy metabolism at multiple levels.

Emboldened by these recent advances, we revisited the fundamental question of ATP set-point regulation. Our central hypothesis to be tested is that mitoflashes might provide the long-sought regulatory mechanism for ATP homeostasis in the mammalian heart. In particular, we sought to determine whether mitoflash activity is able to regulate mitochondrial ATP production, how

mitoflash responds to altered ATP supply and expenditure, and whether manipulation of mitoflashes can reset the level at which the cellular ATP concentration is maintained stable. In the scenario that mitoflash emerges as the ATP set-point regulator, we also attempted to identify a possible physiological trigger that couples the mitoflash activity with the regulation of ATP homeostasis. Our findings indicate that, by sensing and counteracting the ATP supply-and-demand imbalance, mitoflashes appear to constitute a digital auto-regulator for the maintenance of ATP homeostasis in the heart.

## Results

### Mitoflashes negatively regulate ATP production in cardiac mitochondria

To interrogate a possible role of mitoflashes in the regulation of ATP homeostasis, we first determined whether mitoflashes exert any direct effect on mitochondrial metabolic activity and, in particular, ATP production. To circumvent complicated homeostatic regulation at the cellular level, we opted to use isolated cardiac mitochondria freshly prepared from mt-cpYFP-transgenic mouse hearts (*Wang et al., 2008*). While the mitochondrial respiration was supported by the presence of succinate, ADP and phosphate, mitoflash activity was visualized with confocal imaging, and the rate of ATP production was measured with the luciferin luminescence method (*Jouaville et al., 1999*). We found that mitoflash events occurred randomly at a rate of $26 \pm 2$ events / (1000 $\mu m^2 \cdot 100s$) (*Figure 1A and B*) and individual mitoflashes exhibited an average rise time of 7 s, peak amplitude of 0.5 ($\Delta F/F_0$), and duration of 22 s, comparable with those in intact cardiac myocytes (*Figure 1—figure supplement 1*). Treatments using the mitochondria-targeted antioxidants, SS31 and mito-TEMPO, or the mitochondrial permeability transition pore (mPTP) inhibitor cyclosporin A (CsA) decreased the mitoflash frequency by 44%, 31%, and 47% (*Figure 1A and B*), whereas the amplitudes and time courses of mitoflashes were largely unchanged (*Figure 1—figure supplement 1*). Parallel luciferin luminescence measurement showed that, along with the repression of mitoflash activity, the rate of ATP production, assessed as the cumulative ATP formation over a 5 min time window, was enhanced by 25%, 24%, and 22%, respectively (*Figure 1C*). This result suggests that mitoflashes negatively regulate ATP production at the organelle level. In other words, mitoflash activity may also be directly involved in the regulation of energy metabolism, in addition to serving a biomarker of respiratory activity (*Gong et al., 2015*).

### Mitoflash response to altered substrate supply

To investigate whether or not mitoflash activity responds to altered energy metabolism, we first tipped the ATP supply-and-demand balance by altering mitochondrial substrate supply in intact cardiac myocytes. As shown in *Figure 2A*, all metabolites examined, including glucose, free fatty acid and pyruvate, dose-dependently increased the mitoflash frequency, consistent with previous reports (*Fang et al., 2011*; *Pouvreau, 2010*; *Wei et al., 2011*; *Gong et al., 2015*). Strikingly, 10 mM pyruvate, the immediate metabolite that enters mitochondria and feeds the tricarboxylic cycle, elicited the most vigorous mitoflash activity, increasing the mitoflash frequency by 12 folds comparing to 5.6 mM glucose (*Figure 2A*). This stimulatory effect on mitoflash activity was fully reversed upon pyruvate washout, and was largely prevented by α-cyano-4-hydroxycinnamic acid, an inhibitor of the mitochondrial pyruvate transporter (*Figure 2A*). Likewise, a large and robust mitoflash response was obtained in beating hearts under Langendorff perfusion: switching from 5.6 mM glucose to 10 mM pyruvate augmented the mitoflash frequency from $1.2 \pm 0.3$ to $11.5 \pm 2.8$ mitoflashes / (1000 $\mu m^2 \cdot 100s$) (*Figure 2B*).

Parallel measurements revealed that the intracellular ATP content (measured with the luciferin luminescence method) and the ATP/ADP ratio (measured with the indicator PercevalHR [*Tantama et al., 2013*]) exhibited an initial overshoot upon the glucose-to-pyruvate switch and returned to its baseline in about 100 s (*Figure 2C*), indicating a transient escape of homeostatic control of the ATP level. Interestingly, the time-course response of mitoflashes showed that the significant rise of mitoflash activity occurred concomitantly with a decline in the ATP transient (*Figure 2C*). Along with the high activity of mitoflashes at the steady state, the rate of oxygen consumption and the cellular levels of NADH and $FADH_2$ were all increased despite the constancy of ATP (*Figure 2D–2F*), indicative of decreased efficiency of ATP synthesis and substantiating the conclusion that mitoflashes negatively regulate ATP production. Thus, the higher mitoflash activity in response to

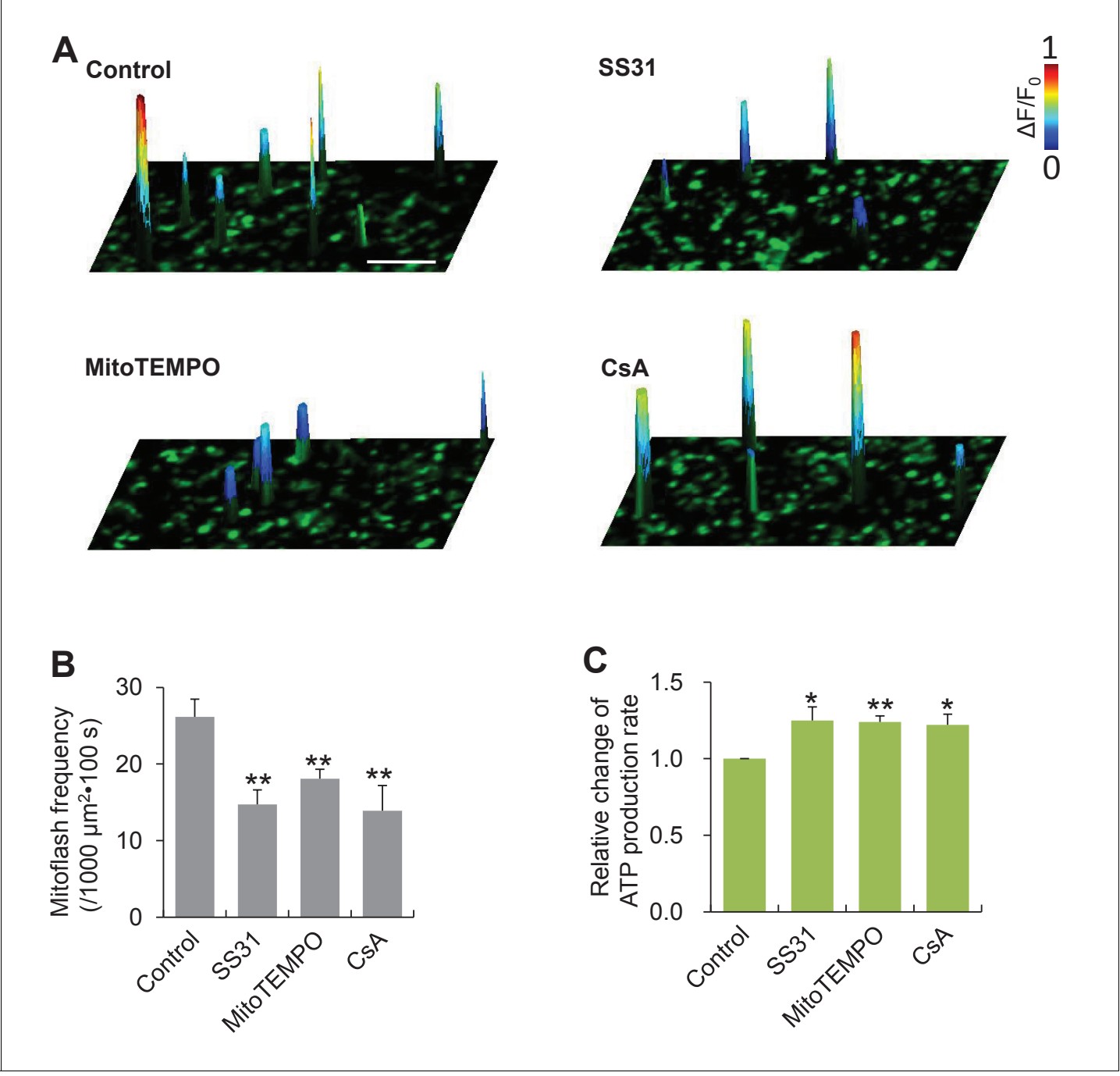

**Figure 1.** Mitoflashes negatively regulate ATP production in isolated cardiac mitochondria. (**A**) Inhibition of mitoflashes by mitochondrial targeted antioxidants, mitoTEMPO (1 µM) or SS31 (50 µM), and the mPTP inhibitor CsA (2 µM). Surface plots show mitoflashes registered in 60 s periods overlaying the respective confocal images. A spike indicates a mitoflash event. Scale bar: 10 µm. (**B**) Statistics of mitoflash frequency. n = 9–22 image files per group; **p<0.01 *versus* control. (**C**) Mitochondrial ATP production measured with the luciferin luminescence assay. Data are expressed as fold-change relative to control group. n = 6 experiments per group; *p<0.05; **p<0.01 *versus* control.

The following source data and figure supplement are available for figure 1:

**Source data 1.** Source data for *Figure 1*.

**Figure supplement 1.** Averaged traces of mitoflashes aligned by onset.

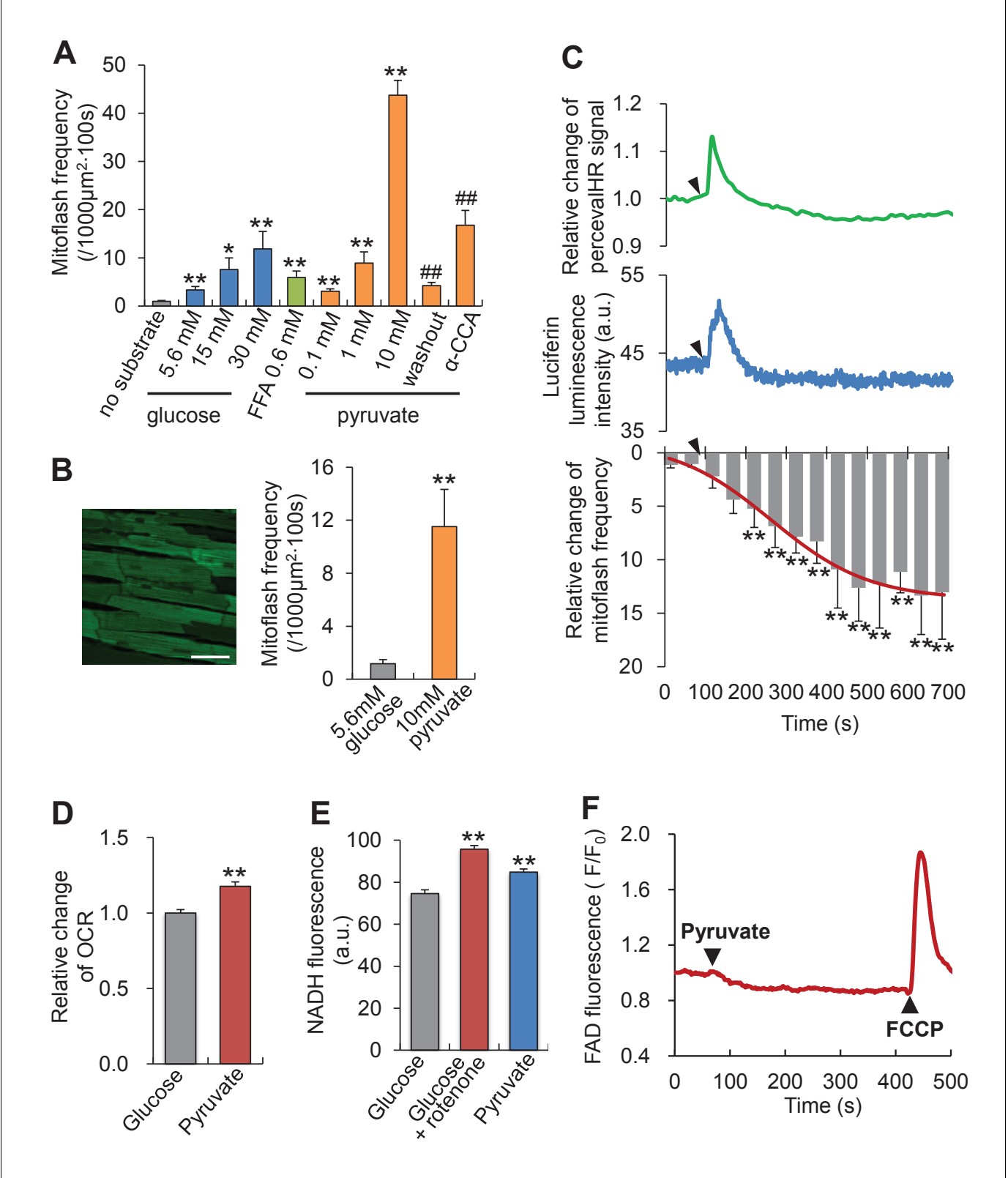

**Figure 2.** Mitoflash and ATP responses to metabolic stimulation in adult cardiac myocytes. (A) Mitoflash frequencies in the presence of different metabolic substrates. n = 14–39 cells per group. *p<0.05; **p<0.01 *versus* substrate-free group; ##p <0.01 *versus* 10 mM pyruvate group. FFA, free fatty-acid. Here we used 0.6 mM palmitate. (B) Pyruvate (10 mM) stimulation of mitoflashes in perfused heart. Left: Confocal image of epimyocardium in an mt-cpYFP transgenic mouse heart under Langendorff perfusion. Scale bar, 50 μm. Right: statistics. n = 13 cells from three hearts. **p<0.01 *versus* 5.6

*Figure 2 continued on next page*

*Figure 2 continued*

mM glucose group. (**C**) Time-resolved ATP/ADP ratio (top, indexed by PercevalHR signal change, n = 39 cells), ATP content (middle, indexed by luciferin luminescence, n = 26 cells), and mitoflash frequency (bottom) in response to pyruvate stimulation (10 mM). Arrow heads indicate the time of adding pyruvate. a.u., arbitrary units. The pH-corrected PercevalHR signal is reported as the normalized fluorescence ratio. The smooth curve overlaying the time-course of the mitoflash frequency was drawn by eye. n = 7 cells per datum point; **p<0.01 *versus* basal conditions. (**D**) Changes of oxygen consumption rate (OCR) upon pyruvate stimulation (10 mM). n = 5 independent experiments. **p<0.01 *versus* 5.6 mM glucose group. (**E**) Changes in NADH content. As a positive control, inhibition of complex I, which oxidizes NADH into $NAD^+$, by 1 μM rotenone increased NADH autofluorescence. n = 54–67 cells per group. **p<0.01 *versus* 5.6 mM glucose group. a.u. arbitrary units. (**F**) FAD autofluorescence upon 10 mM pyruvate stimulation. The trace is the averaged data from 16 cells. Arrow heads indicate the time of adding pyruvate or FCCP. Note that a decrease of FAD indicates an increase of $FADH_2$.

The following source data is available for figure 2:

**Source data 1.** Source data for *Figure 2*.

superfluous substrate supply tends to dampen ATP production and restore the ATP concentration to its set-point.

## Mitoflash response to altered energy expenditure

In the second set of experiments, we aimed to investigate the response of mitoflashes to ATP supply-and-demand imbalance caused by altering energy expenditure. In cardiac myocytes, a single heartbeat consumes ~2% of the cellular ATP (*Jacobus, 1985*). The higher the heart rate, the greater is the rate of ATP expenditure and replenishment. We therefore assessed the mitoflash response to electrical pacing at various frequencies. To minimize interference from cell shortening, image acquisition in contracting cells was synchronized with the electrical pacing (See Materials and methods). The mitoflash activity in pyruvate-treated cells declined precipitously at the onset of electrical pacing, alongside the onset of intracellular $Ca^{2+}$ transients and cell shortenings; it was readily restored at the offset of pacing, with the cessation of $Ca^{2+}$ and contractile responses (*Figure 3A* and *Figure 3—figure supplement 3*). Quantitatively, mitoflash frequency was diminished by 45% at 1 Hz, 69% at 2 Hz, and 80% at 5 Hz pacing (*Figure 3B*). Similar results were found in cardiomyocytes exposed to physiological 5.6 mM glucose in Tyrode's solution, i.e., increasing the workload inhibited rather than stimulated mitoflash activity (*Figure 3C*). Remarkably, the cellular ATP content and ATP/ADP ratio were held constant regardless of pacing at different frequencies (*Figure 3D*), substantiating the tightness of the ATP set-point regulation.

Because the oxidative phosphorylation (OXPHOS) activity is expected to be elevated upon increasing workloads, together with the results in *Figure 2*, it is clear that the mitoflash frequency is not a unique function of mitochondrial respiration. This finding reinforces the notion that mitoflash plays an active role on its own. To this end, we noticed that the rapid and reversible mitigation of mitoflash activity in response to pacing and the inverse relation between mitoflash activity and pacing frequency fit naturally the cell logic that the mitoflash acts as a rapid and sensitive responder to altered ATP demand. For instance, when the ATP demand is augmented at the onset of pacing, the mitoflash activity is downregulated to alleviate its inhibition on ATP production (*Figure 1*), leading to increased ATP supply. Similar reasoning is applicable to explain the equally fast mitoflash changes on cessation of electrical pacing. Taken together, we propose a unifying interpretation - mitoflash acts as a reporter of ATP supply-and-demand imbalance and an ATP homeostasis regulator through negative regulation of ATP production. That mitoflash can serve as the biomarker of mitochondrial respiration is merely a special case when the substrate supply is changed and at the same time the ATP demand is held constant.

Paradoxically, pacing-induced suppression of mitoflashes occurred concurrently with the onset of large cyclic intracellular $Ca^{2+}$ transients, opposite to the prediction of $Ca^{2+}$-dependent activation of mitoflashes in different cell types including cardiac myocytes (*Jian et al., 2014*; *Gong et al., 2014*). This discrepancy is only apparent, because cardiac mitochondria are relatively inert in terms of $Ca^{2+}$ uptake (*Fieni et al., 2012*; *Williams et al., 2013*; *Chen et al., 2011a*) and a prolonged pacing protocol elicited only a moderate mitoflash-stimulatory effect as an after-effect (*Gong et al., 2014*). Thus,

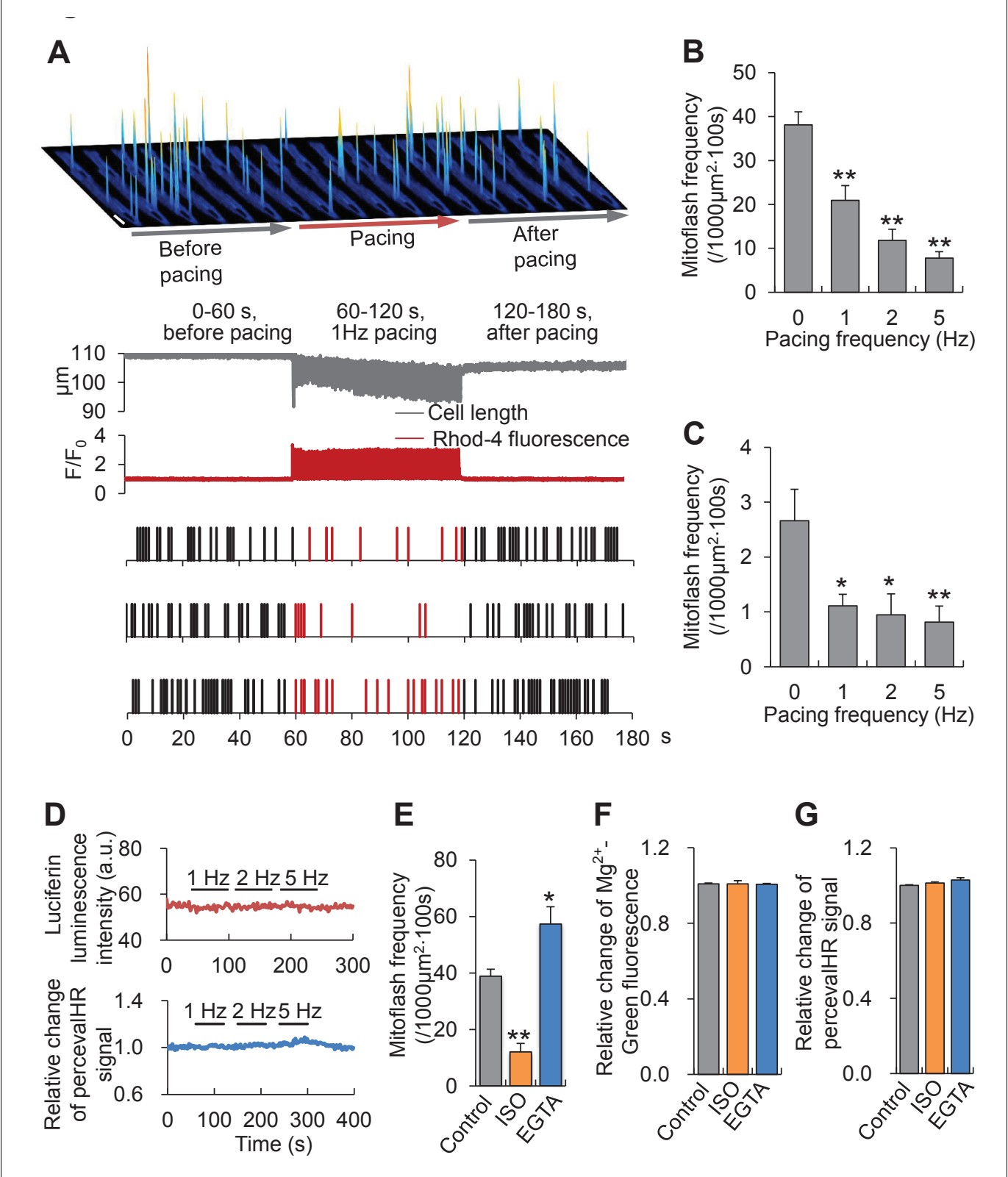

**Figure 3.** Mitoflash and ATP responses to workload changes in adult cardiac myocytes. (**A**) Electrical pacing suppressed mitoflash activity. Top: Space-time plots of mitoflashes right before, during, and right after 1 Hz electrical pacing in a cardiac myocyte in the presence of 10 mM pyruvate. Each plot shows events registered in 10 s periods. Scale bar, 10 µm. Middle: Continuous recordings of cell length and $Ca^{2+}$ transients. Downward and upward deflections indicate cell shortenings and $Ca^{2+}$ transients reported by Rhod-4 fluorescence, respectively. Bottom: Raster plots of mitoflash incidence in

*Figure 3 continued on next page*

*Figure 3 continued*

three representative cells subject to the pacing protocol. (B) Inverse relationship between the incidence of mitoflashes during pacing and the pacing frequency at 10 mM pyruvate. n = 12–17 cells per group. **p<0.01 *versus* resting group. (C) Inverse relationship between the mitoflash frequency during pacing and the pacing frequency at 5.6 mM glucose. n = 8–27 cells per group. *p<0.05; **p<0.01 *versus* resting group. (D) Real-time measurement of cellular ATP content with the luciferin luminescence assay (n = 37 cells) and ATP/ADP ratio with PercevalHR (n = 8 cells) during pacing at different frequencies. a.u., arbitrary units. The pH-corrected PercevalHR signal is reported as the normalized fluorescence ratio. (E–G) Effects of $\beta$-adrenergic stimulation and EGTA treatment on mitoflash activity (E), ATP content (measured with $Mg^{2+}$-Green) (F), and ATP/ADP ratio (G). Isoproterenol (ISO; 1 μM) or 5 mM EGTA in 0 $Ca^{2+}$ Tyrode's solution was used to alter $Ca^{2+}$ cycling and hence cellular energy expenditure in quiescent cardiac myocytes in 10 mM pyruvate. For panel D, n = 13–19 cells per group. *p<0.05; **p<0.01 *versus* control. For panel E, n = 7–15 cells per group. For panel F, n = 6–9 cells per group.

The following source data and figure supplement are available for figure 3:

**Source data 1.** Source data for *Figure 3*.
**Figure supplement 1.** $Ca^{2+}$ transients and cell shortenings elicited by electrical pacing.

the bioenergetics-dependent mechanism might be more powerful than the $Ca^{2+}$-dependent mechanism, thus dominating the mitoflash response in the present experimental conditions.

Next, we extended our studies to other situations of altered ATP demand. Even in quiescence, it takes a significant portion of energy expenditure for a cardiac myocyte to maintain physiological $Ca^{2+}$ gradients across different compartments. In this regard, we demonstrated that $\beta$-adrenergic stimulation with isoproterenol (ISO, 1 μM), to enhance $Ca^{2+}$ cycling and thus to increase ATP demand, inhibited mitoflash frequency by 70% in pyruvate-treated cells (*Figure 3E*). In contrast, brief removal of extracellular $Ca^{2+}$ (5 mM EGTA in nominal 0 $Ca^{2+}$ solution) increased mitoflash activity by 48% (*Figure 3E*). These results provide additional lines of evidence that mitoflash regulation is dominated by bioenergetics - rather than $Ca^{2+}$-dependent mechanism in the present experimental settings. Notably, the ATP content and ATP/ADP ratio were kept constant under either ISO stimulation or $Ca^{2+}$ manipulation (*Figure 3F and G*). Taken together, both sets of experiments unveil a unifying interpretation of the mitoflash responses: mitoflash activity waxes and wanes in accordance with the ATP supply-and-demand imbalance, with higher activity occurring in response to over-exuberant ATP supply or diminished ATP demand, and vice versa.

## Shifting ATP Set-point by altering mitoflash activity

The above results indicate that mitoflashes responding to altered energy supply and demand safeguard the ATP-set-point. The question arises whether tuning mitoflash activity can shift the ATP set-point when both energy expenditure and substrate supply are held unchanged. To answer this question, we inhibited mitoflashes with mitochondria-targeted antioxidants SS31 or mitoTEMPO, or over-expression of SOD2 in intact cardiac myocytes. As shown in *Figure 4A and B*, all these manipulations decreased mitoflash frequency and increased the ATP content even in intact cardiac myocytes. Quantitatively, the ATP set-point was elevated by ~60% while the mitoflash activity was mitigated by ~40% (*Figure 4A and B*). Linear regression revealed a strong inverse correlation between the mitoflash activity and the cellular ATP content (r = −0.96, p=0.009) (*Figure 4B*). A similar trend was seen in cells under physiological condition (at 5.6 mM glucose) displaying a lower basal mitoflash incidence (r = −0.79, p=0.11) (*Figure 4C and D*). Together with the above findings that mitoflashes inhibits ATP production, this result indicates that mitoflash activity is a potent ATP set-point regulator in cardiac myocytes.

## Triggering mitoflashes by proton leakage through ATP synthase

The above results indicate that mitoflashes regulate the cellular ATP homeostasis in a digital and frequency-modulatory manner. Then, what is the specific mechanism that senses the ATP supply-and-demand imbalance and regulates mitoflash activity? To date, three mitochondrial signals have been shown to be effective triggers or activators of mitoflashes, i.e., matrix $Ca^{2+}$, basal ROS, and protons (at nanodomains of the inner mitochondrial membrane) (*Hou et al., 2013*; *Jian et al., 2014*; *Wang et al., 2016b*). Mitochondrial $Ca^{2+}$ is unlikely the candidate signal for the bioenergetics-

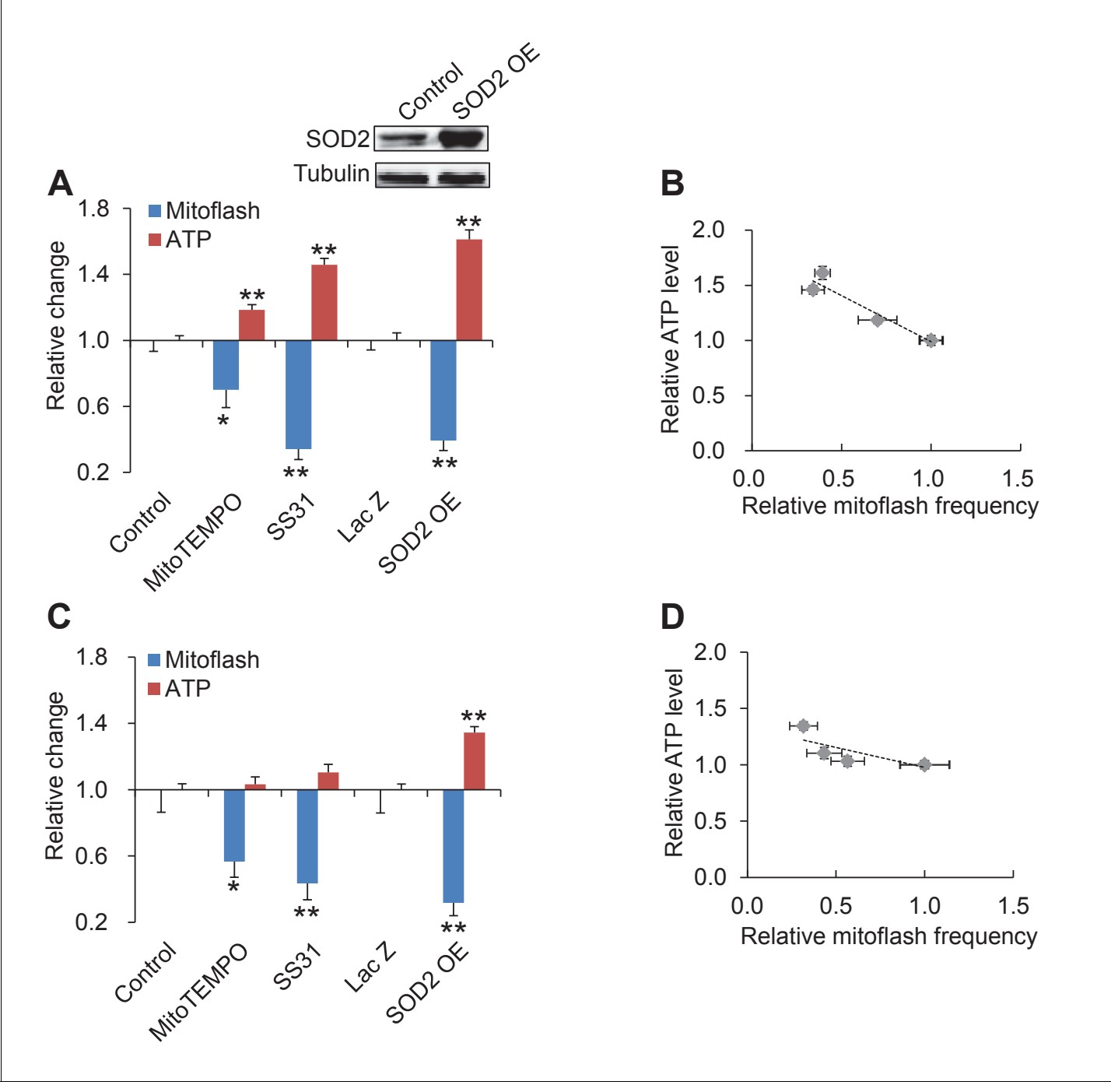

**Figure 4.** Tuning mitoflash frequency shifts cellular atp set-point. (**A**) In the presence of 10 mM pyruvate, mitoflash activity was manipulated by mitoTEMPO (1 μM), SS31 (100 μM), or SOD2 overexpression (SOD2 OE). n = 17–51 cells per group for mitoflash detection. n = 169–488 cells per group for ATP measurement with the luciferin luminescence assay. *p<0.05; **p<0.01 *versus* control. For SOD2 OE, Lac Z overexpression (Lac Z) was used as the control. Data are expressed as fold-change relative to respective control group. Inset shows a representative western-blot of SOD2. (**B**) Inverse relation between ATP content and mitoflash frequency. Dashed line shows the linear regression yielding. r = −0.96, p=0.009. (**C**) As in (**A**), except that data were obtained in 5.6 mM glucose. n = 15–36 cells per group for mitoflash detection and 98–531 cells per group for ATP measurement. *p<0.05; **p<0.01 *versus* control or LacZ. (**D**) Linear regression analysis of (**C**) (r = −0.79, p=0.11).

The following source data is available for figure 4:

**Source data 1.** Source data for *Figure 4*.

dependent mitoflash activity, because no apparent $Ca^{2+}$ changes were detected upon pyruvate-stimulated mitoflash activation (*Figure 5—figure supplement 1A*) and electrical pacing elicited opposite mitoflash and $Ca^{2+}$ responses. Moreover, inhibiting MCU with Ru360 did not show any significant effect on mitoflash frequency (*Figure 5—figure supplement 2*). The lack of significant effect of Ru360 on mitoflashes in cardiomyocytes is expected because pyruvate treatment did not induce any changes in cytosolic $Ca^{2+}$ and thus mitochondrial $Ca^{2+}$ uptake (*Figure 5—figure supplement 1A*). However, mitoflash is sensitive to MCU inhibition in situations when it is induced by elevations of mitochondrial $Ca^{2+}$, e.g., in HeLa cells under hyperosmotic stress or high cytosolic $Ca^{2+}$ stimulation (in permeabilized cells) (*Hou et al., 2013*; *Jian et al., 2014*). Further, electrical pacing did not alter (*Figure 5—figure supplement 1C*) and pyruvate stimulation caused a slight decrease of mitochondrial ROS (*Figure 5—figure supplement 1B*), indicating that ROS are unlikely a major player, either. TMRM measurement revealed little effects of these manipulations on $\Delta\Psi_m$, excluding changes in $\Delta\Psi_m$ as a possible cause of the mitoflash response (*Figure 5—figure supplement 1D*). Given that protons act as a potent trigger of mitoflashes (*Wang et al., 2016b*), we reckoned that ATP synthesis-uncoupled proton entry might be a logic candidate for the trigger of mitoflashes to counteract the ATP supply-and-demand imbalance.

While proton entry at the $F_1F_o$-ATP synthase drives ATP production, there exist multiple routes for ATP synthesis-uncoupled futile proton entry, and varying the proportion of the coupled and uncoupled proton entry would effectively alter the metabolic efficiency. We therefore determined the proton leakage by measuring the ATP-synthesis uncoupled oxygen consumption rate (OCR). Interestingly, we found that the proton leakage associated OCR was augmented by 28% in response to increasing ATP supply, i.e. 10 mM pyruvate stimulation (*Figure 5—figure supplement 3*). Increasing workloads by ISO treatment attenuated the proton leakage associated OCR by 11% while decreasing energy expenditure by brief removal of extracellular $Ca^{2+}$ (5 mM EGTA in nominal 0 $Ca^{2+}$ solution) elevated the proton leakage associated OCR by 14% (*Figure 5—figure supplement 3*). This result indicates that the ATP-synthesis uncoupled proton leakage varies in accordance with the changes of ATP supply and demand, consistent with its role in sensing the ATP supply-and-demand imbalance and triggering mitoflashes.

In search for the specific uncoupled proton leak for the triggering of mitoflashes, we examined possible involvement of proton leakage through mitochondrial uncoupling proteins (UCPs). Inhibition of UCP2, the dominant isoform of UCPs in heart (*Ricquier and Bouillaud, 2000*) (*Figure 5—figure supplement 4*) with either genipin (*Zhang et al., 2006*) or RNA interference, slightly increased, rather than decreased, the mitoflash incidence (*Figure 5A–5C* and *Figure 5—figure supplement 4*). This result excludes UCPs as the source of mitoflash-triggering proton flux. Recently, a Bcl-xL-regulated proton leakage through $F_1F_o$-ATPase has been demonstrated with multiple approaches, including patch-clamping of sub-mitochondrial vesicles, proton flux measurement, and immunochemical demonstration of a Bcl-xL interaction with the $\beta$-subunit of ATP synthase in hippocampal neuronal mitochondria (*Alavian et al., 2011*; *Chen et al., 2011b*). We therefore turned our attention to this $F_1F_o$-ATPase-mediated proton leakage with its hallmark sensitivity to Bcl-xL regulation. Indeed, we found that Bcl-xL overexpression, to inhibit the uncoupled proton flux, markedly decreased mitoflash incidence, whereas knockdown of Bcl-xL with siRNAs (*Figure 5—figure supplement 5*), increased mitoflashes at both 5.6 mM glucose and 10 mM pyruvate (*Figure 5D and E*). Likewise, treatment with ABT-737, a mimetic of BH3-only proteins that inhibits Bcl-xL (*Oltersdorf et al., 2005*), also elevated mitoflash production (*Figure 5D*). Furthermore, the mitoflash responses were accompanied by a 19% upward shift of the ATP set-point with Bcl-xL overexpression, and an 11% downward shift with ABT-737 treatment (*Figure 5F and G*). These lines of evidence support the notion that proton leakage through ATP synthase is an important physiological trigger of mitoflashes, coupling the mitoflash biogenesis with auto-regulation of ATP homeostasis in cardiac myocytes.

## Discussion

### Mitoflashes as the ATP Set-point regulator in the heart

It has been established for decades that myocardial ATP level remains constant in the face of large fluctuations in the rate of ATP consumption and production. Indeed, the present results show that the ATP content and ATP/ADP ratio are held constant regardless of electrical pacing, altered

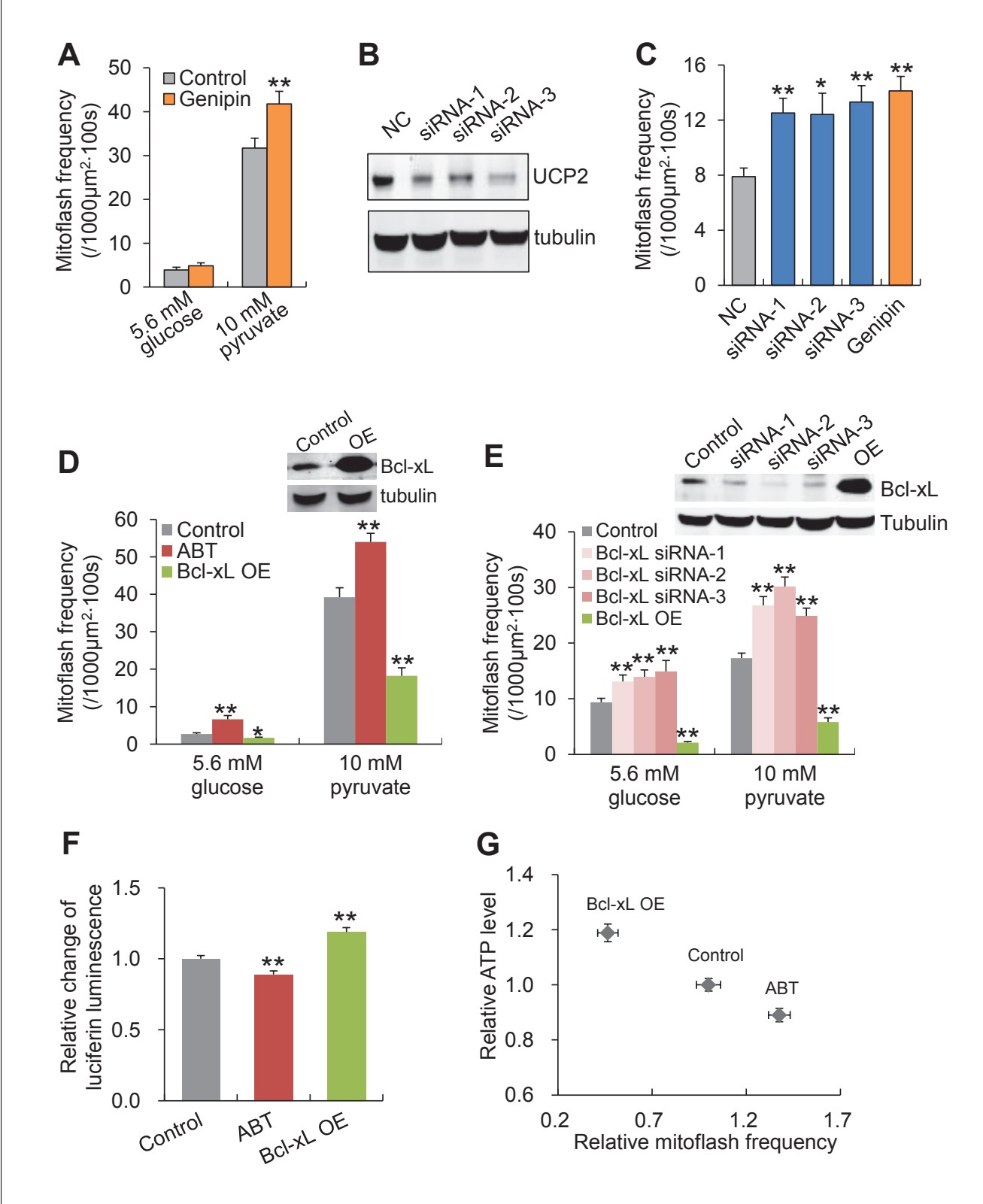

**Figure 5.** Proton leakage through $F_1F_o$-ATPase, not UCP2, triggers mitoflashes. (**A–C**) Effect of UCP2 knockdown and inhibition on mitoflashes. (**A**) Slight change of mitoflash frequency after UCP2 inhibition with genipin (50 µM). n = 12 adult cardiac myocytes per group. \*\*p<0.01 *versus* control. (**B**) Representative western-blot for UCP2 knockdown in neonatal cardiac myocytes. (**C**) Effects of UCP2 knockdown or genipin treatment (50 µM) on mitoflash activity in neonatal cardiac myocytes. n = 28–42 cells per group. \*p<0.05; \*\*p<0.01 *versus* NC. (**D**) Mitoflash activity altered by Bcl-xL

*Figure 5 continued on next page*

*Figure 5 continued*

inhibition or overexpression (OE) in adult cardiac myocytes. Bcl-xL was inhibited by ABT-737 (ABT, 10 μM). Inset shows representative western-blot for Bcl-xL overexpression. n = 33–37 cells per group. *p<0.05; **p<0.01 *versus* control. (E) Activation of mitoflashes by Bcl-xL knockdown in neonatal cardiac myocytes. Three double-strand small RNAs (siRNA-1, siRNA-2, siRNA-3) targeted to Bcl-xL or a negative control siRNA (control) were transfected. Inset shows representative western-blot analysis for Bcl-xL knockdown. n = 22–60 cells per group. *p<0.05; **p<0.01 *versus* control. (F) Effects of Bcl-xL inhibition or overexpression on ATP content. Adult cardiac myocytes were in 10 mM pyruvate containing solution. n = 203–363 cells per group. **p<0.01 *versus* control. (G) Inverse relation between ATP content and mitoflash frequency in the presence of 10 mM pyruvate.

The following source data and figure supplements are available for figure 5:

**Source data 1.** Source data for *Figure 5*.

**Figure supplement 1.** $Ca^{2+}$, basal ros and mitochondrial membrane potential in adult cardiac myocytes responding to pyruvate stimulation or workload alterations.

**Figure supplement 2.** Effect of inhibiting MCU on mitoflashes in neonatal cardiac myocytes.

**Figure supplement 3.** Changes of uncoupled proton leakage by altering ATP supply and demand.

**Figure supplement 4.** Expression of UCP isoforms in cardiac myocytes.

**Figure supplement 5.** Western-blot analysis of Bcl-xL Knockdown.

substrate supply, receptor stimulation, and $Ca^{2+}$ manipulations in isolated single cardiac myocytes, attesting to the exquisiteness of the ATP set-point regulation. Ironically, as ADP, Pi and $Ca^{2+}$ have been dismissed as the major players, the exact mechanism underlying ATP set-point regulation remains an unsolved enigma. From the design principle of control systems, at least three criteria should be met for any candidate mechanism to be qualified as an ATP set-point regulator. First, it should have an impact on mitochondrial ATP production through the OXPHOS process. Second, it should respond to cellular ATP supply-demand imbalance. More specifically, its direction of response should counteract the imbalance and its dynamic range as well as kinetics of response should be robust and fast enough to cope with 10-fold fluctuations of ATP supply and demand on a beat-to-beat basis. Third, actively tuning this regulatory mechanism should be able to reset the ATP level in situations when the ATP supply and demand are held unaltered. Additional mechanism is also required to sense the state of ATP supply-demand imbalance and to couple the regulator's response with the OXPHOS process. In this scenario, the major finding of the present study is that the recently-discovered, digital mitoflash activity meets all these criteria at once.

First, the current study provides direct evidence that mitoflashes negatively regulate ATP production in isolated cardiac mitochondria. Inhibiting mitoflash activity can increase the rate of ATP production by 25%. Multiple mechanisms could contribute to the mitoflash-mediated inhibition of ATP production. First of all, multifaceted energy-consuming processes occur in a mitoflash, including diversion of electrons from energy metabolism to bursting superoxide production, dissipation of $\Delta\Psi_m$, and loss of membrane permeability due to flickering mPTP opening (*Wang et al., 2008*, *2016b*). Albeit the ETC activity is accelerated as indicated by transient matrix alkalization and depletion of the electron-donor pools of NADH and $FADH_2$, $\Delta\mu_H$ for ATP synthesis is dissipated because of loss of $\Delta\Psi_m$. All these events indicate that mitochondria undergoing a mitoflash are at a state of futile respiration. That is, intermittent mitoflashes of the ATP-generating organelles are analogous to discharges of the safety-valve of a working steam engine. Second, as $\Delta\mu_H$ dissipates, the Complex V could operate in the reverse mode, switching on its ATPase activity and consuming ATP in the flashing mitochondria. Third, mitoflashes may activate some yet-unknown downstream pathways to impose prolonged inhibition of OXPHOS activity. Regardless of specific mechanism of action, the finding that mitoflashes negatively regulate ATP production sheds new light on possible bioenergetics role of this digital activity built-in in the powerhouse of the cell.

Second, based on results from numerous maneuvers purported to perturb the ATP supply-and-demand balance, a unifying pattern emerges in which mitoflashes counteract the imbalance between

ATP supply and expenditure, whereby safeguarding the ATP set-point. Specifically, the mitoflash frequency increases with either superfluous substrate supply or diminished energy demand (e.g., offset of electrical pacing, removal of extracellular $Ca^{2+}$); conversely, it mitigates upon substrate washout or with enhanced workload and energy expenditure (e.g., onset of electrical pacing, $\beta$-adrenergic stimulation). Remarkably, the dynamic range of mitoflash frequency spans over an order of magnitude and its kinetics are fast enough to cope with sudden changes occurring in the pacing protocol, allowing for only a transient escape of ATP from its tightly controlled constant level. Thus, compared to the interpretation of mitoflashes as a biomarker of mitochondrial respiration (*Gong et al., 2015*), which holds true in certain conditions (e.g., varying substrate supply without altering the ATP demand), we re-interpret mitoflashes as a reporter of ATP supply-and-demand imbalance and an ATP set-point regulator through its ability to regulate ATP production.

Third, not only that the mitoflash stabilizes the ATP level in fluctuations of energy expenditure and substrate supply, but also it participates in determining the specific level at which cellular ATP is held stable. When the mitoflash activity is manipulated independently of changes in ATP supply and demand, e.g., by ROS scavenging, SOD2 overexpression, and Bcl-xL manipulation, it can tune the ATP set-point upwardly or downwardly: the higher the mitoflash activity is, the lower the ATP set-point becomes. The negative correlation between mitoflash activity and ATP level attained is in remarkable contrast to the constancy of ATP level in situations of electrical pacing, receptor stimulation and calcium manipulation, and testifies the robustness of this mitoflash-mediated ATP set-point regulation.

It is of interest to note that mitoflash regulation of the ATP set-point bears prominent features of a digital, distributed control system. Individual mitoflashes occur only intermittently, and are confined to single mitochondria. They are digital and operate in the frequency-modulatory mode, i.e, the average amplitude and duration of mitoflashes are relatively constant in different conditions. At the single-mitochondrion level, a mitoflash is a dramatic event blanketing the entire organelle and interrupting the ATP synthesis. In a cardiac myocyte typically containing 5,000–8,000 mitochondria, only a tiny portion of mitochondria undergo the flashing activity at any moment; No central control mechanism is identifiable in this control system, because stochastic mitoflash events are evenly distributed among the population of mitochondria, with their frequency rapidly and tightly regulated in accordance with the ATP supply-and-demand imbalance. Keeping in mind that ATP diffuses freely in the cytosol, such local, stochastic ATP-negating events afford a novel mechanism to finely regulate ATP homeostasis at the cellular level. Taken together, we conclude that mitoflashes act as the long-sought ATP set-point regulator in the mammalian heart.

## Bcl-xL-sensitive proton leakage is a physiological trigger of mitoflashes

In search for possible mechanism that couples metabolism and mitoflashes in the context of ATP set-point regulation, we demonstrate that the proton leakage through $F_1F_o$-ATP synthase regulated by Bcl-xL (*Chen et al., 2011b*; *Alavian et al., 2011*) provides a physiological proton trigger of mitoflashes for the regulation of metabolic efficiency. Our recent studies have shown that protons at the nanodomains of inner mitochondrial membrane trigger mitoflashes (*Wang et al., 2016b*), suggesting a possible link between mitoflash biogenesis and the OXPHOS process at the deepest mechanistic level. In the present study, we first show that the ATP synthesis- uncoupled proton leakage varies in accordance with alterations in ATP supply or expenditure, increasing when the supply is superfluous and decreasing when the ATP demand is high. More importantly, we then provide direct evidence that bidirectional manipulations of Bcl-xL-sensitive proton leakage, but not those of UCP2, cause robust, bidirectional changes in the mitoflash frequency and ATP content.

By analogy of local $Ca^{2+}$ signalling (*Cheng and Lederer, 2008*), we have envisaged a 'local control model' for proton triggering of mitoflashes (*Wang et al., 2016b*) in order to explain the finding that protons from different sources differ in terms of mitoflash triggering efficiency. Recent numerical analysis has revealed that free protons are short-lived (lifetime ~1.4 ns) and diffuse only over a nanometre scale (~2.1 nm) in the matrix environment (*Wang et al., 2016b*). Furthermore, free protons are extremely scarce in an alkaline mitochondrial matrix (~0.4 protons in a mitochondrion of 2 μm length and 200 nm diameter at pH 8.0), and matrix pH buffering capacity is estimated to be about 5 mM per pH unit at pH 8.0 (i.e., 1:500,000 for free: bound proton ratio) (*Poburko et al., 2011*). Thus, we envision that only protons in the nanoscopic vicinity of the putative trigger sites can effectively serve their role as the mitoflash trigger. Consistent with this model, super-resolution

microscopy has shown, in the mitochondria from neurons, nanoscale protein partition of UCP4 and $F_1F_o$-ATP synthase, residing in the inner boundary membrane and the cristae membrane, respectively (*Klotzsch et al., 2015*). Several groups have independently proposed that $F_1F_o$-ATP synthase is of the same molecular identity as that of mPTP (*Giorgio et al., 2013*; *Alavian et al., 2014*; *Bonora et al., 2013*; *Bernardi et al., 2015*), whose flickering openings signify the ignition of mitoflashes (*Wang et al., 2008*; *Hou et al., 2014*). If this picture turns out to be true, both proton leakage and mitoflash triggering would occur within the same macrocomplex protein machinery, thus conferring high selectivity among different proton sources.

In summary, we have presented a set of novel and cohesive findings on mitoflashes as a digital auto-regulator of ATP homeostasis in the heart. First, mitoflashes respond rapidly and robustly to ATP supply-and-demand imbalance, increasing with a superfluous supply and mitigating with a greater demand. Because mitoflashes negatively regulate mitochondrial ATP production, such bioenergetics-dependent mitoflash activity acts as an auto-regulator of the ATP homeostasis (mode I, *Figure 6*), akin to the safety-valve of a steam engine. Second, tuning the mitoflash activity is able to shift the level at which ATP is held constant (mode II, *Figure 6*), revealing that mitoflash activity is an important, heretofore unappreciated determinant of the ATP set-point. Third, we demonstrate that the Bcl-xL-sensitive proton leakage through $F_1F_o$-ATP synthase constitutes a physiological proton trigger of mitoflashes (*Figure 6*). These findings not only mark mitoflashes as the putative auto-regulator for cellular ATP homeostasis, but also uncover compelling cell logic for the biogenesis of mitoflashes in the heart.

# Materials and methods

## Reagents

Luciferin, carbonyl cyanide 4-(trifluoromethoxy) phenylhydrazone (FCCP), pyruvate, palmitate, α-cyano-4-hydroxycinnamic acid (α-CCA), isoproterenol (ISO), blebbistatin, cyclosporin A (CsA), antimycin A, and genipin were from Sigma. Ethylene glycol tetraacetic acid (EGTA) was from Amresco. ABT-737 (ABT) was from Selleck Chemicals (Houston, Texas). Rotenone was from Calbiochem.

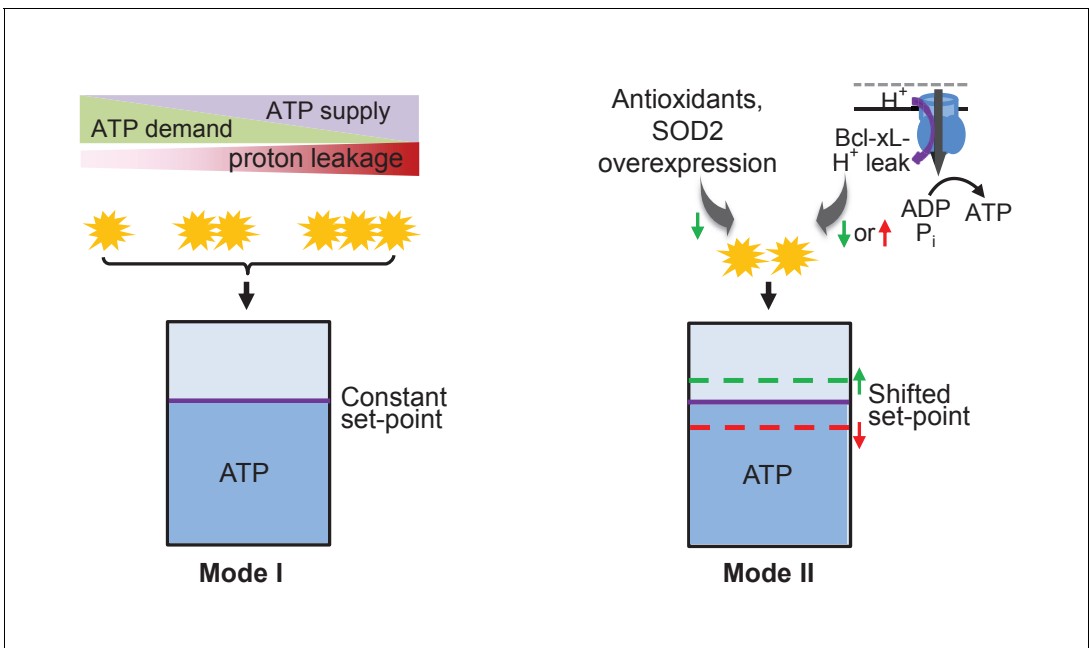

**Figure 6.** Schematic of mitoflash regulation of ATP set-point. In mode I, mitoflash activity, which is triggered by ATP synthesis-uncoupled proton leakage, waxes and wanes in accordance with fluctuations in ATP supply or demand, and whereby stabilizes the ATP set-point at a constant level. In mode II, selective tuning of the mitoflash activity (e.g., Bcl-xL-regulated proton leakage through $F_1F_o$-ATP synthase, or antioxidant repression of mitoflash activity) suffices to shift the ATP set-point upward (at decreased mitoflash activity) or downward (at increased mitoflash activity).

MitoTEMPO was from Enzo Life Sciences. $Mg^{2+}$-Green AM, Rhod-4 AM, BCECF AM, and mitoSOX were from Invitrogen (Eugene, Oregon). The tetrapeptide SS31 (D-arg-dmt-lys-phe-$NH_2$) was synthesized as described (*Zhao et al., 2004*).

## Animal care

All procedures were carried out according to the rules of the American Association for the Accreditation of Laboratory Animal Care International and approved by the Animal Care Committee of Peking University accredited by AAALAC International (IMM-ChengHP-1). This investigation conformed to the Guide for the Care and Use of Laboratory Animals published by the US National Institutes of Health (NIH Publication No. 85–23, revised 1996).

## Adult cardiac myocyte isolation, Culture, and Adenovirus Infection

Single ventricular myocytes were enzymatically isolated from the hearts of adult male Sprague–Dawley rats (200–250 g) or adult C57BL-6 mice (25–35 g), as described previously (*Cheng et al., 1993*; *Shang et al., 2016*). Freshly-isolated cardiac myocytes were plated on culture dishes coated with laminin (Sigma) for 1 hr and then the attached cells were cultured in M199 medium (Invitrogen, Carksbad, California) along with (in mM) 5 creatine, 2 L-carnitine, 5 taurine, and 25 HEPES (all from Sigma). Cells were then infected with adenovirus carrying mt-cpYFP, firefly luciferase, Bcl-xL, or the SOD2 gene at an m.o.i. of 20 and experiments were performed after 60–72 hr in culture.

## Neonatal cardiac myocyte isolation, Culture, Adenovirus Infection, and siRNA Transfection

Ventricular myocytes were isolated from 1-day-old Sprague-Dawley rats, as described previously (*Shen et al., 2007*). Myocytes were plated at $5 \times 10^5$ cells/$cm^2$ in DMEM (Invitrogen) supplemented with 10% FBS (Hyclone) in the presence of 0.1 mM 5-bromo-2-deoxyuridine (Sigma). Adenovirus infection or siRNA transfection was implemented after 24 hr quiescence in serum-free DMEM following 48–72 hr culture in DMEM containing 10% FBS. For adenovirus infection, cells were infected with adenovirus carrying mt-cpYFP, mt-GCaMP5 or the Bcl-xL gene at an m.o.i. of 20. For siRNA transfection, 100 nM siRNA was transiently transfected using Lipofectamine RNAiMax (Invitrogen) according to the manufacturer's instructions. The knockdown efficiency was assessed by western-blot.

## Isolation of cardiac mitochondria

The mitochondria were isolated from mouse hearts as previously reported (*Zhao et al., 2012*). Briefly, mouse hearts were washed with ice-cold isolation buffer (210 mM mannitol, 70 mM sucrose, 5 mM HEPES (pH 7.4), 1 mM EGTA and 0.5 mg/ml BSA), minced, and homogenized. The homogenate was centrifuged at 4°C for 10 min at 1000 g, and the supernatant was collected and further centrifuged at 4°C for 10 min at 12000 g. The pellet was re-suspended for functional assessment.

## Confocal imaging

An inverted confocal microscope (Zeiss LSM 710) with a 40×, 1.3 NA oil-immersion objective was used for imaging. When acquiring the mt-cpYFP signal, images were captured by exciting alternately at 488 and 405 nm, and collecting the emission at >505 nm. For mitoSOX measurement, the indicator (5 μM) was loaded at 37°C for 20 min followed by washing three times. The mitoSOX fluorescence was reported by exciting at 514 nm and collecting the emission at 559–740 nm.

In typical time-series recordings of mitoflashes, 100 frames of 900 × 256 (for adult cardiac myocytes) or 512 × 512 pixels (for neonatal cardiac myocytes) were collected at 0.10–0.14 μm/pixel in bidirectional scanning mode. The frame rate was 30–60 frames/min, and the axial resolution was set to 1.0 μm. All experiments were performed at room temperature (22–26°C) unless specified otherwise.

For ex vivo imaging of mitoflashes, the heart was excised from mt-cpYFP transgenic mice (10–14 weeks old) (*Wang et al., 2008*), and the ascending aorta was cannulated with a customized needle. The heart was perfused in the Langendorff configuration under constant perfusion pressure (~90 mmHg) with oxygenated (100% $O_2$) Tyrode's solution consisting of (in mM) 137 NaCl, 5.4 KCl, 1.2 $MgCl_2$, 1.2 $NaH_2PO_4$, 1.8 $CaCl_2$, 5.6 glucose or 10 pyruvate, and 20 HEPES (pH 7.35, adjusted with

NaOH) at 37°C. The image acquisition plane was focused ~30 µm deep into the epimyocardium and the motion artefacts due to spontaneous beating of the heart were minimized with 10 µM blebbistatin (Sigma).

For imaging mitoflashes in isolated cardiac mitochondria, the mitochondria were suspended in a solution consisting of (in mM) 100 potassium aspartate, 1.0 $MgCl_2$, 20 KCl, 0.5 EGTA, 10 glutathione, 20 HEPES, 8% dextran (MW 35,000–45,000), 10 $KH_2PO_4$, 2.5 succinate, and 0.2 ADP (pH 7.2). Typically, 100 frames of 512 × 512 pixels were collected at the rate of 60 frames/min.

## Mitoflash detection in contracting cells

In a subset of experiments, electrical field stimulation (5 ms square-wave pulses at 2× threshold voltage) was applied *via* a pair of platinum electrodes, and cells were perfused with Tyrode's solution at a rate of 3 ml/min. Cells displaying healthy contractility were chosen for mitoflash measurement and those near the anode, where electrolysis can produce reactive oxygen species locally (*Jackson et al., 1986*), were avoided. These cautious measures were necessary because we have recently shown that ROS generated by electrolysis of Tyrode's solution at the anode can cause significant increase in mitoflash as well as mitochondrial $Ca^{2+}$ (*Zhang et al., 2014*). To minimize interference from cell shortening, image acquisition in contracting cells was at 1 s/frame and synchronized with the electrical pacing, such that it was phase-locked with cell contraction. In parallel experiments, cell length and cytosolic $Ca^{2+}$ transients were monitored continuously by placing the scan line along the long axis of the cell. The cytosolic $Ca^{2+}$ transients were measured by Rhod-4 with excitation at 543 nm and emission collection at 548–646 nm. Fluorescent images of the line were acquired every 3.78 ms and the time-course of cell shortening was extracted from the line-scan images by an edge-detection algorithm.

## Real-time measurement of ATP content and ATP/ADP Ratio

Cellular ATP content was measured directly with the firefly luciferase-catalysed chemiluminescence method (*Jouaville et al., 1999*) and indirectly with the small-molecule indicator $Mg^{2+}$-Green (*Campanella et al., 2008*). For chemiluminescence measurement, cultured cardiac myocytes expressing firefly luciferase were bathed in Tyrode's solution with 1 mM luciferin (Sigma). The luminescence was recorded with an electron-multiplying charge-coupled device camera (Andor iXon DU-897D-C00-#BV, Andor Technology, South Windsor, CT) and analysed by AndoriQ software (version 1.0). To measure the ATP content with $Mg^{2+}$-Green, cells were loaded with 5 µM $Mg^{2+}$-Green AM and the fluorescence was recorded with excitation at 488 nm and emission at >560 nm.

The cellular ATP/ADP ratio was measured by expressing PercevalHR (*Tantama et al., 2013*) in cardiac myocytes. Briefly, the PercevalHR fluorescence was acquired by alternate excitation at 488 nm and 405 nm and emission collection at 506–702 nm. To remove the pH bias of PercevalHR, we adopted the procedure developed previously (*Tantama et al., 2013*). Specifically, the changes of BCECF fluorescence and PercevalHR fluorescence upon pH alterations were recorded and an empirical linear correlation between the PercevalHR signal and the BCECF signal was established for calibration. The pH bias in the PercevalHR signal was corrected by subtracting the estimated pH component based on the pH calibration of PercevalHR and parallel BCECF measurement.

For measuring ATP production in isolated cardiac mitochondria, the isolated mitochondria were suspended in respiration solution containing (in mM) 220 mannitol, 70 sucrose, 5 $KH_2PO_4$, 2.5 $MgCl_2$, 0.5 EDTA, 2.5 succinate, 0.2 ADP, 2 HEPES (pH7.4), and 0.1% BSA. The ATP content was measured with the luciferase assay (Promega, Madison, Wisconsin).

## Measurement of Oxygen Consumption Rate

Oxygen consumption rate in intact adult cardiac myocytes was measured with a Clark-type oxygen electrode (Strathkelvin 782 2-Channel Oxygen System version 1.0; Strathkelvin Instruments, Motherwell, UK). Briefly, 1 × $10^5$ cells were suspended in Tyrode's solution containing 5.6 mM glucose or 10 mM pyruvate and the oxygen consumption was measured over 10 min with the Strathkelvin System.

To measure ATP synthesis-uncoupled oxygen consumption rate, the mouse cardiac myocytes were cultured in XF24 cell-culture microplates (Seahorse Bioscience). The cells were suspended in different medium: Tyrode's solution with 5.6 mM glucose, Tyrode's solution with 10 mM pyruvate,

Tyrode's solution with 10 mM pyruvate and 1 µM ISO, nominal 0 $Ca^{2+}$ Tyrode's solution with 10 mM pyruvate and 5 mM EGTA. Bioenergetics analyses were performed in an XF24 Extracellular Flux Analyzer (Seahorse Bioscience) with the injection of oligomycin (1 µM), FCCP (1 µM), and rotenone (1 µM) and antimycin A (1 µM) sequentially. The uncoupled oxygen consumption rate was calculated as the oligomycin-insensitive percentage relative to basal oxygen consumption.

## Western-blot analysis

Cell lysates were separated by 4–12% NuePAGE (Invitrogen) and transferred to nitrocellulose membranes (Millipore). Membranes were blocked with 5% non-fat dry milk and incubated with primary antibody overnight at 4°C. Monoclonal anti-UCP2 antibody (Santa Cruz Biotechnology, California), monoclonal anti-Bcl-xL (Sigma), polyclonal anti-SOD2 (Abcam, RRID:AB_300434), and anti-α-tubulin antibody (Sigma, RRID:AB_477593) were used. Blots were visualized using secondary antibodies conjugated with IRDye (LI-COR, Lincoln, Nebraska) and an Odyssey imaging system (LI-COR).

## Quantitative Real-time PCR

Total RNA was extracted from adult or neonatal cardiac myocytes using Trizol reagent (Invitrogen) and converted to cDNA using M-MLV reverse transcriptase (TaKaRa, Japan) with oligo-dT primers. Quantitative real-time PCR was performed using Trans Start Green qPCR Super Mix (TransGen Biotech, China) with primers designed with the NCBI/Primer-BLAST tool (see the following table for the primer sequences). The Ct values were obtained using CFX Manager Quantification software (BioRad, Hercules, California) and the relative expression of target genes was analysed by comparison with 18S rRNA.

## Image processing and mitoflash analysis

Confocal images were analysed using custom-developed programs written in Interactive Data Language (IDL, ITT). Cell-motion artefacts and background fluorescence changes were corrected by image processing and individual mitoflashes were located with the aid of FlashSniper (*Li et al., 2012*; *LJHIS007, 2017*; https://github.com/ljhis007/flashsniper. A copy is archived at https://github.com/elifesciences-publications/flashsniper).

## Statistics

Data are expressed as mean ± SEM. When appropriate, Student's t-test was applied to determine the statistical significance, and a simple linear regression model was used to investigate the correlation between mitoflash activity and ATP content. The coefficient of determination ($R^2$) was used to evaluate the goodness of fit of the model. Prism linear regression analysis was used for the statistical analysis. $p < 0.05$ was considered statistically significant. The number of samples chosen for each comparison was determined based on past similar experiments.

## Acknowledgements

We thank Drs SS Liu, W Wang, MQ Dong, RP Xiao, YM Wang, JC Luo, M Ouyang, and CQ Song for valuable comments, X Wang for the generous gift of SOD2 adenovirus, T Ye for the synthesis of SS31, and IC Bruce and WL Yan for manuscript editing. This work was supported by the National Key Basic Research Program of China (2013CB531200) and the National Science Foundation of China (31130067, 31470811, and 31521062).

## Additional information

### Funding

| Funder | Grant reference number | Author |
|---|---|---|
| National Science Foundation of China | 31130067 | Heping Cheng |
| National Key Basic Research Program of China | 2013CB531200 | Heping Cheng |

| National Science Foundation of China | 31470811 | Xianhua Wang |
| National Science Foundation of China | 31521062 | Heping Cheng |

The funders had no role in study design, data collection and interpretation, or the decision to submit the work for publication.

## Author contributions

XW, Conceptualization, Data curation, Formal analysis, Supervision, Funding acquisition, Validation, Investigation, Visualization, Methodology, Writing—original draft, Writing—review and editing; XZ, Data curation, Formal analysis, Validation, Investigation, Methodology, Writing—original draft; DW, Data curation, Formal analysis, Validation, Investigation, Methodology; ZH, Data curation, Formal analysis, Validation, Investigation; TH, CJ, WQ, Data curation, Validation, Investigation, Methodology; PY, FL, RZ, Data curation, Investigation, Methodology; TS, Software, Validation, Methodology; JL, Data curation, Software, Validation, Methodology; YW, Resources, Data curation, Investigation, Methodology; FG, Supervision, Validation, Writing—review and editing; HC, Conceptualization, Supervision, Funding acquisition, Investigation, Methodology, Writing—original draft, Writing—review and editing

## Author ORCIDs

Xianhua Wang, http://orcid.org/0000-0002-2016-9415

## Ethics

Animal experimentation: All procedures were carried out according to the rules of the American Association for the Accreditation of Laboratory Animal Care International and approved by the Animal Care Committee of Peking University accredited by AAALAC International (IMM-ChengHP-1). This investigation conformed to the Guide for the Care and Use of Laboratory Animals published by the US National Institutes of Health (NIH Publication No. 85-23, revised 1996).

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
