## [Decision Letter]

Thank you for submitting your article "Mitochondrial Flashes Regulate ATP Homeostasis in the Heart" for consideration by *eLife*. Your article has been favorably evaluated by Marianne Bronner (Senior Editor) and three reviewers, one of whom is a member of our Board of Reviewing Editors.

The reviewers have discussed the reviews with one another and the Reviewing Editor has drafted this decision to help you prepare a revised submission.

This study investigates the role of mitochondrial flashes in regulating the ATP set-point in the cardiac muscle cells. The results show that the frequency of mitoflashes is negatively correlated with the ATP production and serves as a key regulator for maintaining the balance of ATP supply-demand. The results show that the BCl^-^xL-regulated proton leakage through F1Fo-ATP synthase appears to mediate the coupling between mitoflash production and the ATP set-point regulation. It is concluded that mitoflashes constitute a digital auto-regulator for ATP homeostasis in the heart.

This is a very important study because it is bringing in a novel concept to explain one of most fundamental questions in bioenergetics, namely, regulation of ATP set-point in beating heart. The authors have applied different approaches to validate their hypothesis. However, there are several major concerns as listed below that either require further explanations or additional experiments. While the authors provide evidence that mito flashes vary according to the interventions applied, the authors have not provided convincing evidence that the mito flashes are setting the ATP homeostasis. Rather mito flashes may be a consequence of alterations in proton flow or proton leak rather than a regulator of ATP production. Furthermore, mito flashes do not occur in every mitochondria suggesting that mito flashes are not playing a significant role in ATP homeostasis. In addition, further experimentation is need to understand the role that calcium plays in frequency of mitoflashes.

1) The ATP measurements in the isolated mitochondria were done in mitochondria suspensions, however, the mitoflash events occurred randomly in only a very small subset of mitochondria (26 ± 2 events / 1000 µm2·100s). The justification for correlating these two measurements was not explained (whole populations of mitochondria vs. a relatively small number of mitochondria).

2) Figure 2 shows the correlation of ATP/ADP, ATP, and flashes. As indicated, glucose-to-pyruvate switch leads to an initial overshoot of mitoflash frequency that coincides with a transient increase of luciferin luminescence (ATP increase), which does not indicate a clear "negative" correlation. Please clarify.

3) Figure 3 used super-high levels of pyruvate, which already significantly increased (many folds) flashes (shown in Figure 2). It is therefore not clear how the pacing induced suppression of this very high level of flashes has any physiological relevance.

4) Also in Figure 3, manipulations that significantly suppressed the super-high level of flashes did not change ATP levels. This does not support the correlation?

5) Figure 4 is confusing, since the manipulations of flashes through antioxidants may exert other effects to alter the detection/measurement of ATP using the fluorescent indicator. The authors need to rule out that the negative correlation was not a result of indirect effect or off-target effect.

6) Figure 5 is the key data to support this study's conclusion. The idea that BCl^-^XL coupled but not UCP coupled proton leak triggers flashes (and suppressed ATP) is interesting. However, the authors need to show with direct measurements that manipulating UCP or BCl^-^XL changes the pH in the microdomain in the matrix near the inner membrane of mitochondria in cardiomyocytes. They also need to explain why and how the leaked protons between these two mechanisms could have a different effect in triggering flashes.

7) What happens to mito membrane potential during the interventions applied and when mitoflashes appear?

8) What happens to the frequency of mito flashes after application of the ATP synthase inhibitor oligomycin?

9) The authors argue that calcium does not regulate the appearance of mito flashes. What effect does inhibition of MCU have on mito flashes? Why does pyruvate which feeds the TCA cycle (a calcium dependent process) increase mito flashes while a calcium chelator also increases mito flashes by 48% (subsection “Mitoflash Response to Altered Energy Expenditure”, last paragraph; Figure 3)?

10) If the authors argue that mitoflashes regulate ATP, what are the circumstances that "mitoflashes are biomarkers of mito respiration" as mentioned by the authors in the manuscript and argued by others? Why different roles?

11) Since mitoflashes do not occur in all mitos is this a consequence of alterations in mito biogenesis? Do they occur in the mitos undergoing autophagy or undergoing fission?

12) The authors state…"Remarkably, the cellular ATP content and ATP/ADP ratio were held constant regardless of pacing at different frequencies (Figure 3), substantiating the tightness of the ATP set-point regulation" but this statement seems in contradiction of mito flashes setting the ATP level if pacing results in a decrease in mito flashes. Please clarify.

[Editors' note: further revisions were requested prior to acceptance, as described below.]

Thank you for resubmitting your work entitled "Mitochondrial Flashes Regulate ATP Homeostasis in the Heart" for further consideration at *eLife*. Your revised article has been favorably evaluated by Marianne Bronner (Senior Editor), a Reviewing Editor, and two reviewers.

The manuscript has been improved but there are some remaining issues that need to be addressed before acceptance, as outlined below:

The authors have made a good effort in addressing the previous concerns. Overall, this study deals with a fundamental question regarding the ATP set-point regulation in beating heart. The data indeed give a strong correlation that mitoflash could be a digital auto-regulator for ATP homeostasis in the heart. Before publication, however, there are still issues concerning the physiological relevance and the causal relationships. There exist several key discrepancies that will require the authors' further consideration. In addition, one reviewer may have overlooked the technical issue last time for flash measurement *during* pacing (see below). It is candidly felt that the authors should not over interpret their data.

1) The results in Figure 5—figure supplement 2 showing that inhibiting MCU with Ru360 did not show any significant effect on mitoflash frequency, which are in contradictory with two recent publications by several key authors in this manuscript. They have previously shown that blocking mitochondrial Ca^2+^ transport by knockdown of MICU1 or MCU, markedly diminished the flash response (PMID: 23283965), consistent with the idea that Ca^2+^ triggers mPTP, the underlying mechanisms for mitoflash. In addition, they have also shown that inhibition of MCU activity by ruthenium red (RR) or by genetic disruption with siRNAs, both significantly depressed superoxide flashes induced by μM [Ca^2+^]c in permeabilized cells. Specifically, RR decreased flash frequency by 73.8%, siMCU-1 by 73.2% and siMCU-2 by 49.1%. These results show that mitoflashes are activated by the steady-state mitochondrial Ca^2+^ elevation (PMID: 24699914). These crucial discrepancies between the authors own data need to be explained in detail and unequivocally.

2) The pacing mediated decreases in flash frequency are quite interesting and surprising. However, it is in opposition to the results published by Dr. Wang's group in University of Washington (PMID: 25252178). As I can see, the only difference between these two reports is that the current mitoflash data are acquired before, during and after pacing. The frequency during pacing is decreased, and it returned back right after the pacing stopped. This is different from Wang's JMCC paper, in which the flash is measured before and after pacing, but not *during* pacing. My concern is that a motion artifact (as shown in their Figure 3 the cell continues to beat, shorten, change volume) could affect the recording of mitoflash. This technical issue needs to be addressed. This technical issue will require the authors to perform simple new experiments by recording the mitoflash right before and after the pacing.

3) I am still bothered by the use of 10 mM pyruvate in the majority of the experiments. I understand that this manipulation increases mitoflash frequency approximately 12 times more than when the cells are incubated in the physiological glucose concentrations (Figure 2), as such, the authors can have a robust mitoflash signal to study. However, this raises the concern about the physiological relevance and the causal relationship of the reported data. This is an important issue because the "artificially" high mitoflash data support a correlation rather than a causal relationship. Please discuss this issue.

4) Another issue, as pointed out in the previous Question#1 that the measurement of ATP (steady state) and flash are on different populations. In fact, the unchanged ATP level during pacing is understandable (Figure 3). What really matters is the flux (the rate of ATP production). The flux (ATP generation rate) is likely increased when the pacing rates are increased, but this is not determined. For instance, it is well-known that without a dynamic and continuous ATP regeneration, in less than 60 beats, the entire cellular ATP reservoir will be consumed. Therefore, pacing the heart will drop the cellular ATP to a minimal level within a few seconds until the Ca^2+^-mediated generation of NADH starts to kick in (see Figure 2 of Brandes R & Bers DM, Biophys J, 2002, PMID: 12124250). Therefore, it is hard to use the steady state ATP measurement to show that ATP flux is controlled by flash mechanistically. Please address this issue.

5) The BCl^-^xL-mediated leak channel proposed by Jonas group (Alavian et al., 2011, Chen et al., 2011) has a peak conductance of 600 pS on average, which should be much higher than the H^+^ leak channel that is proposed here. Subsequent studies by Dr. Jonas group suggest this channel could be mPTP itself.

---

## [Author Response]

*[…] This is a very important study because it is bringing in a novel concept to explain one of most fundamental questions in bioenergetics, namely, regulation of ATP set-point in beating heart. The authors have applied different approaches to validate their hypothesis. However, there are several major concerns as listed below that either require further explanations or additional experiments. While the authors provide evidence that mito flashes vary according to the interventions applied, the authors have not provided convincing evidence that the mito flashes are setting the ATP homeostasis. Rather mito flashes may be a consequence of alterations in proton flow or proton leak rather than a regulator of ATP production. Furthermore, mito flashes do not occur in every mitochondria suggesting that mito flashes are not playing a significant role in ATP homeostasis. In addition, further experimentation is need to understand the role that calcium plays in frequency of mitoflashes.*

In the revised manuscript, we have performed additional experiments and added further explanations accordingly. In particular, 1) we inhibited the activity of mitochondrial Ca^2+^ uniporter (MCU) with Ru360 and found that the MCU inhibition did not exert any significant effect on mitoflash activity (Figure 5—figure supplement 2); 2) we examined the effect of electrical pacing and cell contraction on mitoflash activity at normal glucose concentration (5.6 mM glucose) and found similar results as those at 10 mM pyruvate – increasing workload inhibits rather than stimulates mitoflash activity (Figure 3—figure supplement 2); 3) we measured mitochondrial membrane potential during different experimental manipulations used in this study (Figure 5—figure supplement 1); 4) we examined the effect of inhibiting ATPase with oligomycin on mitoflash activity and validated that SS31 and mitoTEMPO did not exert any side-effect on ATP measurement. In addition, we discussed how mitoflashes might negate ATP production: possible mechanisms proposed include electrochemical energy dissipation, complex V operating as ATPase during a mitoflash, and possible prolonged inhibitory effects on oxidative phosphorylation (OXPHOS) activity.

Several lines of evidence indicate that mitoflash constitutes an important regulator of ATP homeostasis in the heart. First, instead of being a bystander or an epiphenomenon of mitochondrial respiration or proton leakage, mitoflashes significantly inhibit ATP production in isolated mitochondria. Second, when the cellular ATP supply-and-demand balance is perturbed, the mitoflash response tends to counteract the imbalance and restore the ATP set-point. Third, when ATP supply and demand are held constant, selectively inhibiting mitoflash activity shifts the otherwise very stable ATP set-point, and quantitatively, there is a negative correlation between mitoflash activity and ATP set-point. Finally, although mitoflashes may not occur in every mitochondrion in a given time window, their ATP-negating effects should be non-local because ATP diffuses freely in the cytosol. We believe these new data and text revisions have significantly improved this work.

*1) The ATP measurements in the isolated mitochondria were done in mitochondria suspensions, however, the mitoflash events occurred randomly in only a very small subset of mitochondria (26 ± 2 events / 1000 µm2·100s). The justification for correlating these two measurements was not explained (whole populations of mitochondria vs. a relatively small number of mitochondria).*

In our experiments, ATP production was measured in the whole mitochondrial population over a 5-min time window after adding the substrate and ADP. We estimated that ~25% mitochondria of the population undergo mitoflashes during this time window (the mitochondrial radius is estimated to be 1 µm). Multiple mechanisms could contribute to the inhibitory effect of mitoflashes on ATP production. First of all, multifaceted energy-consuming processes occur in a mitoflash, including diversion of electrons from energy metabolism to bursting superoxide production, dissipation of ΔΨm, and loss of membrane permeability due to flickering mPTP opening. All these events indicate that mitochondria during a mitoflash are at a state of futile respiration, consuming NADH, FADH_2_, and electrochemical energy. Second, when ΔμH dissipates, flashing mitochondria may directly consume ATP through Complex V operating as an ATPase. Moreover, mitoflashes may activate some yet-unknown downstream pathways for prolonged negative regulation of OXPHOS activity. It is noteworthy that, although individual mitoflashes are spatially confined, ATP is highly diffusible. As such, mitoflashes afford a mechanism to finely regulate the ATP supply-and-demand balance at the cellular level. These points have been clarified in the revised manuscript.

*2) Figure 2 shows the correlation of ATP/ADP, ATP, and flashes. As indicated, glucose-to-pyruvate switch leads to an initial overshoot of mitoflash frequency that coincides with a transient increase of luciferin luminescence (ATP increase), which does not indicate a clear "negative" correlation. Please clarify.*

The negative correlation between changes in mitoflash frequency and ATP level is applicable when ATP supply and demand are held constant, and when the changes reach steady state (Mode II in Figure 6). In Figure 2, the initial overshoot reflects a transient “escape” of homeostatic control of the ATP level. The decline of ATP from its peak appears to reflect that the mitoflash regulatory mechanism kicked in to restore the ATP level to its set-point (Mode I in Figure 6). We have clarified this point in the revised manuscript.

*3) Figure 3 used super-high levels of pyruvate, which already significantly increased (many folds) flashes (shown in Figure 2). It is therefore not clear how the pacing induced suppression of this very high level of flashes has any physiological relevance.*

Per your suggestion, we performed the pacing experiment at 5.6 mM glucose and confirmed that increasing workload inhibits rather than stimulates mitoflash activity. The new data have been added in the revised manuscript (Figure 3—figure supplement 2).

*4) Also in Figure 3, manipulations that significantly suppressed the super-high level of flashes did not change ATP levels. This does not support the correlation?*

When the ATP supply and demand are unchanged, there exists a negative correlation between mitoflash activity and ATP levels (Mode II in Figure 6). When the ATP supply-and-demand balance is perturbed by altering either ATP supply or workload, the mitoflash response counteracts the imbalance (Mode I in Figure 6). In Figure 3, we show that the mitoflashes are largely suppressed upon enhancing the workloads induced by pacing stimulation. Under this condition, the ATP expenditure is highly increased due to the imbalanced ATP supply-and-demand that might lead to ATP downward shift. However, the rapid decrease of mitoflash activity to alleviate its inhibition on ATP production counteracts this imbalance and maintains the cellular ATP at constant levels. Thus, the results in Figure 3 are consistent with the scenario presented in Mode I in Figure 6.

*5) Figure 4 is confusing, since the manipulations of flashes through antioxidants may exert other effects to alter the detection/measurement of ATP using the fluorescent indicator. The authors need to rule out that the negative correlation was not a result of indirect effect or off-target effect.*

We tested the indirect or side- effect of SS31 and mitoTEMPO on ATP measurement using the in vitroluciferin luminescence assay. As shown in Figure 7, neither SS31 nor mitoTEMPO exhibited any significant effect on ATP measurement.

Author response image 1.Effect of SS31 or mitoTEMPO on ATP measurement.10 nM and 50 nM ATP standards were used in the luciferin luminescence assay with or without adding SS31 (100 μM) or mitoTEMPO (1 μM). n = 4 independent experiments.**DOI:**
http://dx.doi.org/10.7554/eLife.23908.021

*6) Figure 5 is the key data to support this study's conclusion. The idea that BCl^-^XL coupled but not UCP coupled proton leak triggers flashes (and suppressed ATP) is interesting. However, the authors need to show with direct measurements that manipulating UCP or BCl^-^XL changes the pH in the microdomain in the matrix near the inner membrane of mitochondria in cardiomyocytes. They also need to explain why and how the leaked protons between these two mechanisms could have a different effect in triggering flashes.*

We thank the reviewer for this insightful comment. Unfortunately, we cannot directly measure the microdomain pH now because there are no proper pH indicators which can be localized to specific microdomain in mitochondrial matrix. However, we have recently performed numerical analyses on local proton signaling in the mitochondria. We have shown that the free protons are short-lived (lifetime ~1.4 ns) and diffuse only over a nanometer scale (~2.1 nm) in the matrix environment (Wang et al., 2016). Furthermore, free protons are extremely scarce in an alkaline mitochondrial matrix (~0.4 protons in a mitochondrion of 2 μm length and 200 nm diameter at pH 8.0), and matrix pH buffering capacity is estimated to be about 5 mM per pH unit at pH 8.0 (i.e., 1:500,000 for free: bound proton ratio) (Poburko et al., 2011). Thus, only when a proton is in nanoscopic vicinity of a putative trigger site, it can potentially trigger a mitoflash. We have included this discussion in the revised manuscript.

*7) What happens to mito membrane potential during the interventions applied and when mitoflashes appear?*

We have measured the mitochondrial membrane potential by interventions used in this study, including pyruvate stimulation, electrical pacing, ISO or EGTA treatment. We found that all these manipulations exert negligible effects on mitochondrial membrane potential. These results are now reported in Figure 5—figure supplement 1. As we and others have reported previously (Wang et al., 2008, Wei-LaPierre et al., 2013, Wang et al., 2016), the mitochondrial membrane potential is transiently depolarized when a mitoflash appears. Further, there is a positive correlation between the amplitudes of mitoflashes and the magnitudes of mitochondrial depolarizations (Figure 1 and Figure S3 in (Wang et al., 2016)).

*8) What happens to the frequency of mito flashes after application of the ATP synthase inhibitor oligomycin?*

Figure 8 shows that the mitoflashes stimulated by 10 mM pyruvate are sensitive to the inhibition of the ATP synthase with oligomycin, in general agreement with what has been seen at 5.6 mM glucose condition (Wang et al., 2008).

Author response image 2.Inhibition of mitoflashes by oligomycin.The adult cardiomyocytes were stimulated by 10 mM pyruvate. 5 μM oligomycin were applied to treat the cells for 30 min before mitoflash detection. n = 12-22 cells for either group. ** p <0.01 versuscontrol.**DOI:**
http://dx.doi.org/10.7554/eLife.23908.022

*9) The authors argue that calcium does not regulate the appearance of mito flashes. What effect does inhibition of MCU have on mito flashes? Why does pyruvate which feeds the TCA cycle (a calcium dependent process) increase mito flashes while a calcium chelator also increases mito flashes by 48% (subsection “Mitoflash Response to Altered Energy Expenditure”, last paragraph; Figure 3)?*

To address your concerns, we examined the effect of inhibiting MCU with Ru360 on mitoflashes. In adult cardiac myocytes, the MCU activity is known to be very low (Fieni et al., 2012, Williams et al., 2013) and Ru360 treatment did not exert any detectable effect on mitochondrial Ca^2+^ uptake during caffeine-induced Ca^2+^ releases from the internal store. In neonatal cardiac myocytes whose MCU activity is relatively high, we found that the mitoflash frequency was unaffected albeit the amplitude of mitochondrial Ca^2+^ transient stimulated by 10 mM caffeine is significantly decreased. The new data are reported in the revised manuscript (Figure 5—figure supplement 2). This result suggests that mitochondrial Ca^2+^ is unlikely a major trigger of mitoflash activation in cardiac myocytes. In contrast, the bioenergetics-dependent mechanism is much more powerful, thus dominating the mitoflash response in the present experimental conditions. Therefore, not only pyruvate which feeds the TCA cycle and stimulates mitochondrial metabolism to increase ATP supply augments mitoflashes, but also a calcium chelator EGTA which decreases ATP demands also increases mitoflashes – both can be unified in a single working model.

*10) If the authors argue that mitoflashes regulate ATP, what are the circumstances that "mitoflashes are biomarkers of mito respiration" as mentioned by the authors in the manuscript and argued by others? Why different roles?*

Mitoflashes can serve as a biomarker of mitochondrial respiration when the substrate supply is changed and at the same time the ATP demand is held constant, such as pyruvate stimulation. Notably, this is merely a special case of a unifying interpretation that we proposed here – mitoflash acts as a reporter of ATP supply-and-demand imbalance and an ATP set-point regulator through inhibition of ATP production. We have clarified this issue in the revised manuscript.

*11) Since mitoflashes do not occur in all mitos is this a consequence of alterations in mito biogenesis? Do they occur in the mitos undergoing autophagy or undergoing fission?*

Because the mitoflash observation time is limited (we usually detect mitoflash in 100 s), we can only detect mitoflashes in a subset of mitochondria. However, mitoflashes do occur stochastically and uniformly across the cell, and all mitochondria are expected to give rise to mitoflashes providing that the detection time is long enough. On your other comment, whether mitoflashes play any roles in mitochondrial biogenesis, autophagy, and fusion-and-fission are interesting questions that merit future investigations. However, they are perhaps beyond the scope of the current study. Thank you!

*12) The authors state…"Remarkably, the cellular ATP content and ATP/ADP ratio were held constant regardless of pacing at different frequencies (Figure 3), substantiating the tightness of the ATP set-point regulation" but this statement seems in contradiction of mito flashes setting the ATP level if pacing results in a decrease in mito flashes. Please clarify.*

As in our response to point 4, under the condition as shown in Figure 3, the increased ATP expenditure might lead to an imbalanced ATP supply-and-demand. This is counteracted by a rapid decrease of mitoflash activity to alleviate its inhibition on ATP production, which tends to maintain the cellular ATP at constant levels. However, when the ATP supply and demand are unaltered, selectively changing mitoflash activity re-sets the ATP level. These two modes of behaviors of the same homeostatic control system are summarized in Figure 6 and our statement is not in contradiction with the roles of mitoflashes in setting and maintaining homeostatic ATP level.

[Editors' note: further revisions were requested prior to acceptance, as described below.]

*The manuscript has been improved but there are some remaining issues that need to be addressed before acceptance, as outlined below:*

*The authors have made a good effort in addressing the previous concerns. Overall, this study deals with a fundamental question regarding the ATP set-point regulation in beating heart. The data indeed give a strong correlation that mitoflash could be a digital auto-regulator for ATP homeostasis in the heart. Before publication, however, there are still issues concerning the physiological relevance and the causal relationships. There exist several key discrepancies that will require the authors' further consideration. In addition, one reviewer may have overlooked the technical issue last time for flash measurement during pacing (see below). It is candidly felt that the authors should not over interpret their data.*

We thank again the editor and the reviewers for additional comments and suggestions. In the revised manuscript, we have clarified the physiological relevance and further explained how to avoid motion artifacts during pacing. In addition, we have toned down our conclusion for the role of mitoflashes in regulating ATP production throughout the Abstract, Introduction, Results and Discussion, to avoid over interpretation of our data.

*1) The results in Figure 5—figure supplement 2 showing that inhibiting MCU with Ru360 did not show any significant effect on mitoflash frequency, which are in contradictory with two recent publications by several key authors in this manuscript. They have previously shown that blocking mitochondrial Ca^2+^ transport by knockdown of MICU1 or MCU, markedly diminished the flash response (PMID: 23283965), consistent with the idea that Ca^2+^ triggers mPTP, the underlying mechanisms for mitoflash. In addition, they have also shown that inhibition of MCU activity by ruthenium red (RR) or by genetic disruption with siRNAs, both significantly depressed superoxide flashes induced by μM [Ca^2+^]c in permeabilized cells. Specifically, RR decreased flash frequency by 73.8%, siMCU-1 by 73.2% and siMCU-2 by 49.1%. These results show that mitoflashes are activated by the steady-state mitochondrial Ca^2+^ elevation (PMID: 24699914). These crucial discrepancies between the authors own data need to be explained in detail and unequivocally.*

The differences between the present results and our previous publications cited are *not* contradictory. Rather, they arise from different experimental conditions in different cell types. In the publication of PMID:23283965, HeLa cells were stimulated by hyperosmotic stress which induced cytosolic Ca^2+^ transient and MCU-sensitive mitochondrial Ca^2+^ uptake. In the publication of PMID: 24699914, the permeabilized HeLa cells were stimulated by high cytosolic Ca^2+^ which also induced MCU-sensitive mitochondrial Ca^2+^ uptake. So inhibiting MCU with ruthenium red or by siRNA knockdown decreased mitochondrial Ca^2+^ uptake and thus repressed mitoflashes. However, in the present study, cardiomyocytes stimulated by 10 mM pyruvate did not

exhibit any detectable changes in cytosolic Ca^2+^ and thus mitochondrial Ca^2+^ uptake (Figure 5—figure supplement 1). So it is not unexpected that inhibiting MCU with Ru360 exerted no effect on mitoflash activity in the present experimental conditions. These points are now integrated in the revised manuscript (subsection “Triggering Mitoflashes by Proton Leakage through ATP Synthase”, first paragraph).

*2) The pacing mediated decreases in flash frequency are quite interesting and surprising. However, it is in opposition to the results published by Dr. Wang's group in University of Washington (PMID: 25252178). As I can see, the only difference between these two reports is that the current mitoflash data are acquired before, during and after pacing. The frequency during pacing is decreased, and it returned back right after the pacing stopped. This is different from Wang's JMCC paper, in which the flash is measured before and after pacing, but not during pacing. My concern is that a motion artifact (as shown in their Figure 3 the cell continues to beat, shorten, change volume) could affect the recording of mitoflash. This technical issue needs to be addressed. This technical issue will require the authors to perform simple new experiments by recording the mitoflash right before and after the pacing.*

We thank the reviewer for this insightful question. As we explained in the Methods, and shown in Figure 3, we did our detection of mitoflashes right before, during, and right after the pacing, and found that the mitoflash activity declined precipitously upon pacing and restored as pacing stopped. This could not be due to motion artifact, because our image acquisition was synchronized to electrical stimulation (i.e., confocal scanning was triggered by the electric stimulator’s SYNC signal such that cell contraction was phase-locked with image acquisition). To make it clearer, we have slightly modified the legend of Figure 3 and the related Methods (subsection “Mitoflash Detection in Contracting Cells”).

In Wang’s JMCC paper, the mitoflash frequency during pacing was not determined and the mitoflash frequency was only modestly increased after 2-min continuous pacing at 2 Hz, along with elevated mitochondrial basal Ca^2+^ in some mitochondria. However, we did not detect any significant changes in mitoflash after 1-min, pacing at 1-Hz. As we stated in the Methods, cells in our experiments “were perfused with Tyrode’s solution at a rate of 3 ml/min. Cells displaying healthy contractility were chosen for mitoflash measurement and those near the anode, where electrolysis can produce reactive oxygen species locally (Jackson et al., 1986), were avoided.” These cautious measures are deemed necessary, because we have recently shown that ROS generated by electrolysis of Tyrode’s solution at the anode can cause significant increase in mitoflash as well as mitochondrial Ca^2+^ (PMID: 24912881). In this regard, we could not find pertinent cell perfusion and selection information in the work by Wang’s group, and therefore could not determine the exact source of discrepancy (if any).

*3) I am still bothered by the use of 10 mM pyruvate in the majority of the experiments. I understand that this manipulation increases mitoflash frequency approximately 12 times more than when the cells are incubated in the physiological glucose concentrations (Figure 2), as such, the authors can have a robust mitoflash signal to study. However, this raises the concern about the physiological relevance and the causal relationship of the reported data. This is an important issue because the "artificially" high mitoflash data support a correlation rather than a causal relationship. Please discuss this issue.*

In the revised manuscript, we have emphasized that the key experiments were all repeated under physiological conditions (5.6 mM glucose), and qualitatively similar results were obtained both at 10 mM pyruvate and 5.6 mM glucose, including changes of mitoflash frequency during pacing at different frequencies (now Figure 3, previously Figure 3—figure supplement 2), inverse relation between mitoflash frequency and ATP level (now Figure 4, previously Figure 4—figure supplement 1), and changes of mitoflashes by manipulating BCl^-^xL (Figure 5).

*4) Another issue, as pointed out in the previous Question#1 that the measurement of ATP (steady state) and flash are on different populations. In fact, the unchanged ATP level during pacing is understandable (Figure 3). What really matters is the flux (the rate of ATP production). The flux (ATP generation rate) is likely increased when the pacing rates are increased, but this is not determined. For instance, it is well-known that without a dynamic and continuous ATP regeneration, in less than 60 beats, the entire cellular ATP reservoir will be consumed. Therefore, pacing the heart will drop the cellular ATP to a minimal level within a few seconds until the Ca^2+^-mediated generation of NADH starts to kick in (see Figure 2 of Brandes R & Bers DM, Biophys J, 2002, PMID: 12124250). Therefore, it is hard to use the steady state ATP measurement to show that ATP flux is controlled by flash mechanistically. Please address this issue.*

We thank the reviewer for the insightful comment. We agree that the steady state of ATP cannot represent the ATP flux. To determine a possible role of mitoflash in ATP production, we used isolated cardiac mitochondria supported by succinate, ADP, and Pi. In the isolated mitochondria system, there is no cellular homeostatic regulation system, and the ATP level reached in a given period of time under this condition represents the *ATP production rate*. The negative correlation between ATP production and mitoflash frequency in isolated mitochondria suggests a regulatory role of mitoflashes in ATP production (flux in vitro).

Per your suggestion, we have toned down our conclusion for the role of mitoflashes in regulating ATP production throughout the Abstract, Introduction, Results, and Discussion, to avoid over interpretation of our data.

*5) The BCl^-^xL-mediated leak channel proposed by Jonas group (Alavian et al., 2011, Chen et al., 2011) has a peak conductance of 600 pS on average, which should be much higher than the H^+^ leak channel that is proposed here. Subsequent studies by Dr. Jonas group suggest this channel could be mPTP itself.*

As you point out, classic proton-selective channels usually are of tiny conductance, while the BCl^-^xL-mediated leak channel proposed by Jonas group (Alavian et al., 2011, Chen et al., 2011) exhibits a peak conductance of 600 ps when recorded in symmetric 120 mM K^+^ at pH 7.3. Of this conductance, its *fractional contribution* by protons is not yet determined. That this leak channel can indeed permeate protons has been established by the experiment showing that ATP-driven H^+^ ion sequestration into sub-mitochondrial vesicles is attenuated by BCl^-^xL inhibitors (their Figure 5). In the case that leak channel itself is also the effector mPTP, proton triggering of the mPTP (bridging the proton gradient across the inner mitochondrial membrane) may operate in a fashion very similar to calcium triggering of the ryanodine receptor (bridging the

calcium gradient across the endo/sarcoplasmic reticulum membrane).